

SciPost Phys. Lect. Notes 13 (2020)

# Aspects of high energy scattering

**Christopher D. White**⋆

Centre for Research in String Theory, School of Physics and Astronomy,
Queen Mary University of London, 327 Mile End Road, London E1 4NS, UK

⋆ christopher.white@qmul.ac.uk

## Abstract

Scattering amplitudes in quantum field theories are of widespread interest, due to a large number of theoretical and phenomenological applications. Much is known about the possible behaviour of amplitudes, that is independent of the details of the underlying theory. This knowledge is often neglected in modern QFT courses, and the aim of these notes - aimed at graduate students - is to redress this. We review the possible singularities that amplitudes can have, before examining the generic behaviour that can arise in the high-energy limit. Finally, we illustrate the results using examples from QCD and gravity.

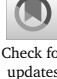
# 1 Motivation

Much research in contemporary theoretical high energy physics involves scattering amplitudes (see e.g. refs. [1, 2] for recent reviews), which are themselves related to the probabilities for interactions between various objects to occur. Usually, this is within the context of a specific field or string theory, and we might therefore be interested in the following questions:

- *What physical behaviour occurs in a given theory?* This might be a known or proposed theory at the LHC, for example, in which case we would want to know what to look for in experiments. In more formal theories (such as $\mathcal{N} = 4$ Super-Yang-Mills), we might be trying to find out what types of behaviour are possible, and how this is similar or different to more applied theories. In gravity, we might be testing the limits of our understanding (e.g. by probing scattering above the Planck scale).

- *What mathematical structures can amplitudes contain?* Recent work has revealed interesting connections between amplitudes and pure mathematics (e.g. number theory, special functions, abstract algebra) [3]. Knowing the types of mathematical object or function that generically occur in amplitudes can help us to address open mathematical problems, or provide shortcuts to calculating amplitudes in the first place (see e.g. [4–9]).

- *Are there special kinematic regimes where calculations become simpler?* It might be possible to gain insights into amplitudes at all orders in perturbation theory, for special regimes of energy, transverse momentum etc. Such ideas have great practical importance. For example, we must often sum certain contributions to all orders in perturbation theory in order to obtain meaningful results for comparison to data [10, 11]. More formally, special kinematic regimes have a key role to play in elucidating the complete behaviour of a theory, overlapping with the first two questions.

Lately, people have also become interested in how different theories can be related to each other. For example, string and field theories can be related by taking certain limits, or through the AdS / CFT correspondence [12]. There are certain string theories, such as *ambitwistor string theory* [13–15], that encode the behaviour of field theories in a string-like language. Furthermore, there are relationships between different types of field theory. Chief amongst these is perhaps the *double copy* [16–18] which, together with similar relationships, relates scattering amplitudes [19] and classical solutions [20–38] in a wide variety of theories including biadjoint scalar, non-abelian gauge and gravity theories, with and without supersymmetry. With this in mind, we might ponder the following:

- *Can we find common languages, that make e.g. QCD and gravity look the same?* Even though the physics in different theories can vary greatly, it might be possible to interpret this physics through a common calculational procedure, that itself makes clear why the physical behaviour is forced to be different.

- *Are there generic behaviours, that any physical theory has to obey?* If we know that certain terms *must* appear in perturbation theory, we are no longer surprised when they do, and indeed know to be on the look-out for them!

The aim of these lectures is to explore these issues within a particular context, namely the high energy, or *Regge limit* [1], to be defined more carefully in what follows. This is important in QCD, given that present-day collider experiments (e.g. the LHC) probe kinematic regimes in which additional contributions from the high energy limit - that go beyond fixed orders

---

[1]Classic reviews of the Regge limit can be found in [39–41]. Reference [42] brings these nicely up to date.

SciPost Phys. Lect. Notes 13 (2020)

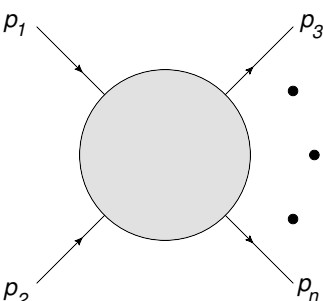

Figure 1: A scattering process, involving particles with 4-momenta $\{p_i\}$.

in perturbation theory - are believed to be important [43–64]. In gravity, the Regge limit is relevant for the scattering of black holes, or other high energy objects. This is of interest due to the recent discovery of gravitational waves by LIGO, but also for studies which aim to explore the possible existence and structure of a gravitational S-matrix if we go above the Planck scale (see e.g. ref. [65]).

In fact, study of the Regge limit goes back to the 1960s [2]. This was part of a large body of work that attempted to construct and understand scattering amplitudes from general principles alone, and which is loosely known as *S-matrix theory*. The programme was particularly important for strong interactions, given that the relevant field equations (QCD) were not yet known. Nor was it clear that there would ever be a perturbative description of strong interactions, given that asymptotic freedom was not known about. Some of the conclusions of S-matrix theory remain highly useful and relevant. For example, one can show that at high energy, amplitudes have similar overall behaviour, regardless of the underlying theory. This allows us to interpret modern results, clarify their structure, and also provides important cross-checks for new calculations.

The structure of these notes is as follows. In section 2, we will review the basics of S-matrix theory, by reviewing the definition of the S-matrix, and outlining the assumptions that go into its characterisation. We will then examine a particular consequence of these assumptions, namely that singularities in the complex energy plane are related to physical bound states and thresholds. In section 3, we look at the Regge limit, and deduce that amplitudes must generically grow in a power-like way as the energy becomes large, where this growth can be associated with the exchange of an infinite family of bound states. In section 4, we illustrate this with examples in QCD and gravity. Although the behaviour in those two theories will be very different, the calculations themselves will be very similar, and both agree with the generic predictions that come from S-matrix theory.

## 2 Scattering and the S-matrix

### 2.1 The S-matrix

In this section, we review the definition of the S-matrix. This material will overlap with what you may have seen in previous courses on quantum field theory. However, it is important to bear in mind that we are trying to be as general as possible here, without assuming any particular underlying theory. We will use the term "particle" to describe those objects which are colliding with each other, but we should not take this to mean that these particles are in any way associated with fields. Furthermore, much of what we say would also apply to other

---

[2]Strictly, speaking, Regge's original paper was in 1959, although this was in a non-relativistic context!

types of colliding object, such as strings or branes (the latter, of course, were not known about in the 1960s!).

We have to start somewhere, so let us assume that we can talk about a scattering process such as that shown in figure 1, which we can split into three different regimes. At very early times ($t \to -\infty$), we have well-defined incoming particles, here with 4-momenta $p_1$ and $p_2$. At very late times ($t \to \infty$), we have a number of well-defined outgoing particles, with 4-momenta $\{p_3, \ldots, p_n\}$. In between these two extremes, some sort of interaction happens, and we would like to know how to systematically describe it. First, however, we should clarify what we mean by "well-defined" incoming and outgoing particles. This typically means that the interaction is short-range, so that the incoming and outgoing particles can be considered as free. We can then expect to describe all possible sets of incoming particles by some set of states $\{|n; \text{in}\rangle\}$, and outgoing particles by a different set of states $\{|m; \text{out}\rangle\}$. Here the integers $n$ and $m$ label the numbers of particles, which is not itself sufficient to fully characterise each state, although this will be a useful short-hand notation in what follows. To fully characterise the states, we need to know the momenta of the particles, and also any additional quantum numbers such as charge and spin. Here and throughout, we shall ignore the complications of spin etc., and simply label states by the 3-momenta of the particles e.g.

$$|n; \text{in}\rangle \equiv |\mathbf{p}_1, \mathbf{p}_2, \ldots, \mathbf{p}_n; \text{in}\rangle,$$

(similarly for outgoing particle states). Note that the 3-momenta are indeed sufficient: if the incoming / outgoing particles are free, then they must obey the relativistic relation

$$p_i^2 = E_i^2 - \mathbf{p}_i^2 = m_i^2,$$

where $E_i$ and $m_i$ are the energy and mass respectively. We can choose to normalise these states however we like, given that in any formula for probabilities to transition between states, we must divide by the normalisation. As you will have seen in your field theory courses, it is conventional to normalise one-particle states according to the convention

$$\left\langle \mathbf{p}_i; \begin{smallmatrix} \text{in} \\ \text{out} \end{smallmatrix} \middle| \mathbf{p}_j; \begin{smallmatrix} \text{in} \\ \text{out} \end{smallmatrix} \right\rangle = (2\pi)^3 2E_i \delta^3(\mathbf{p}_i - \mathbf{p}_j). \tag{1}$$

Here the delta function indicates that states with different momenta are orthogonal (i.e. particles with different momenta are physically different!). The element of choice affects the prefactors, which here contain an overall power of $(2\pi)$, and the energy, to be explained shortly. We have also indicated in eq. (1) that the same normalisation criterion holds either for incoming or outgoing particle states. It is straightforward to generalise eq. (1) to two arbitrary multiparticle states:

$$\left\langle \mathbf{p}_1, \mathbf{p}_2, \ldots, \mathbf{p}_n; \begin{smallmatrix} \text{in} \\ \text{out} \end{smallmatrix} \middle| \mathbf{p}_1', \mathbf{p}_2', \ldots \mathbf{p}_m'; \begin{smallmatrix} \text{in} \\ \text{out} \end{smallmatrix} \right\rangle = \delta_{nm} \prod_{i=1}^n (2\pi)^3 2E_i \delta^3(\mathbf{p}_i - \mathbf{p}_i'). \tag{2}$$

Here we see in particular that states with different numbers of particles are physically distinct, and thus orthogonal. We will assume that the states are *complete*, namely that any state in the (Fock) space we are working in can be represented as a superposition of the possible particle states. More formally, this is expressed by the completeness relation

$$\sum \left| n; \begin{smallmatrix} \text{in} \\ \text{out} \end{smallmatrix} \right\rangle \left\langle n; \begin{smallmatrix} \text{in} \\ \text{out} \end{smallmatrix} \right| = 1, \tag{3}$$

where the right-hand side denotes the identity operator, and the sum on the left-hand side is over all possible particle numbers, and also their momenta. More fully, this can be written out as

$$\sum_{n=0}^\infty \left( \prod_{i=1}^n \int \frac{d^3\mathbf{p}_i}{(2\pi)^3 2E_i} \right) \left| \mathbf{p}_1, \ldots, \mathbf{p}_n; \begin{smallmatrix} \text{in} \\ \text{out} \end{smallmatrix} \right\rangle \left\langle \mathbf{p}_1, \ldots, \mathbf{p}_n; \begin{smallmatrix} \text{in} \\ \text{out} \end{smallmatrix} \right| = 1. \tag{4}$$

The denominator in the integral over the particle momenta arises from the normalisation of the states in eq. (1), as can be easily checked by acting on a particular particle state with eq. (4). The reason for this normalisation is that the measure of the momentum integral can be shown to be Lorentz invariant by itself, which is very convenient when constructing formulae for cross-sections etc. (see your QFT courses!). If we assume that the set of possible outgoing particles is the same as the set of possible incoming ones, it must be true that the in and out states defined above form a basis of the *same* free theory. Thus, they must be relatable. To make this concrete, we can define the operator

$$\hat{S} = \sum_m |m; \text{in}\rangle\langle m; \text{out}|, \tag{5}$$

and orthogonality of the basis states then implies that

$$\hat{S}|n; \text{out}\rangle = |n; \text{in}\rangle, \tag{6}$$

so that the $\hat{S}$ operator "turns out states into in states". We can further show that

$$\langle n; \text{in}|\hat{S}|m; \text{in}\rangle = \langle n; \text{out}|m; \text{in}\rangle = \langle n; \text{out}|\hat{S}|m; \text{out}\rangle, \tag{7}$$

namely that matrix elements of the $\hat{S}$ operator do not depend on whether we take the in or out states as our basis. We can arrive at a physical interpretation of $\hat{S}$ by looking at the middle of eq. (7), and remembering that the probability to transition from some initial state $|m; \text{in}\rangle$ to some final state $|n; \text{out}\rangle$ is given, via the usual rules of quantum mechanics, by

$$P_{nm} \propto |\langle n; \text{out}|m; \text{in}\rangle|^2. \tag{8}$$

Thus, squared matrix elements of the $\hat{S}$ operator are related to the probability for a given set of particles to scatter to another. For this reason, $\hat{S}$ is called the *scattering operator*, and scattering probabilities are related to *elements of the S-matrix*

$$S_{nm} = \left\langle n; \begin{smallmatrix} \text{in} \\ \text{out} \end{smallmatrix} \left| \hat{S} \right| m; \begin{smallmatrix} \text{in} \\ \text{out} \end{smallmatrix} \right\rangle. \tag{9}$$

Given that it relates in and out states, it must encode all properties of the interaction. If we are able to write down a complete set of particle states, and can also give all possible elements of the S-matrix, we have completely described what a given theory can do, at least when it comes to scattering. Given that eq. (9) does not care whether we use the in or out states as our basis (as long as we choose the same states on each side of the matrix element!), we can omit the explicit in / out notation in what follows, unless otherwise stated.

## 2.2 Properties of the S-matrix

The previous section reviewed material that you are probably familiar with from your quantum field theory courses. However, we stress again that we have at no point assumed a particular underlying theory - merely the existence of quantum particles that can scatter. In doing so, and in saying that the in and out states could be related to each other, we have assumed the *superposition principle* of quantum mechanics. Furthermore, we assumed that the incoming and outgoing particles were described by states in a free (non-interacting) theory, so that the interactions themselves were *short-range*. By making these and more properties explicit, we can examine the consequences for the S-matrix in a systematic way. We will see that we can surmise quite a lot about what scattering looks like on generic grounds, subject to a very reasonable set of assumptions.

As well as superposition and short range interactions, we will further assume that our theory is consistent with special relativity, so that the S-matrix must be *Lorentz invariant* [3]. In practice, this means that elements of the S-matrix must depend only on scalar products of 4-vectors. Another important property follows from completeness of the states. From the definition of the scattering operator in eq. (5), one finds

$$\hat{S}^{\dagger} = \sum_m |m; \text{out}\rangle\langle m; \text{in}|, \tag{10}$$

and one may then use eq. (3) to show that

$$\hat{S}^{\dagger}\hat{S} = \hat{S}\hat{S}^{\dagger} = 1. \tag{11}$$

In words, we say that the S-matrix is *unitary*, but we stress again that this is merely a consequence of the fact that we assumed that our particle states are complete: given any two complete bases of a Fock space, any transformation that relates them must be unitary. Furthermore, the physical interpretation of unitarity is that probability is conserved in scattering processes. This is again a statement about the completeness of the states: if there was a complete set of in states, for example, but *not* a complete set of out states, it would be possible for information to "get lost" when we scatter our particles. This would then show up as a failure to conserve probability, although it is difficult to see how such a theory makes any sense at all, let alone how such a consequence could ever be testable.

Let us write the unitarity condition in more detail. We can first rewrite the measure for momentum integration that we encountered in eq. (4), using the well-known identity (see e.g. ref. [66])

$$\int \frac{d^3\mathbf{p}_i}{(2\pi)^3 2E_i} = \int \frac{d^4 p_i}{(2\pi)^3} \delta^{(+)}(p_i^2 - m_i^2). \tag{12}$$

Here we have introduced

$$\delta^{(+)}(p_i^2 - m_i^2) = \delta(p_i^2 - m_i^2)\theta(p_i^0), \tag{13}$$

where $\delta(x)$ and $\theta(x)$ are the Dirac delta and Heaviside functions respectively. Physically, eq. (13) tells us that the particle whose momentum we are summing over must be on-shell (i.e. obey the relativistic energy-momentum relation $p_i^2 = m_i^2$), and have positive energy ($p_i^0 > 0$). Using eq. (12), the unitarity condition of eq. (11), sandwiched between given initial and final states with momenta $\{p_i\}$ and $\{p_i'\}$ respectively, can be written as

$$\sum_{n=0}^{\infty} \left( \prod_{i=1}^{n} \frac{d^4 q_i}{(2\pi)^3} \delta^{(+)}(q_i^2 - m_i^2) \right) \langle \mathbf{p}_1', \ldots, \mathbf{p}_{m'}' | \hat{S} | \mathbf{q}_1, \ldots, \mathbf{q}_n \rangle \langle \mathbf{q}_1, \ldots, \mathbf{q}_n | \hat{S}^{\dagger} | \mathbf{p}_1, \ldots, \mathbf{p}_m \rangle$$
$$= \langle \mathbf{p}_1', \ldots, \mathbf{p}_m' | \mathbf{p}_1, \ldots, \mathbf{p}_m \rangle, \tag{14}$$

where we have inserted a complete set of states with 4-momenta $\{q_i\}$. Equation (14) is going to be enormously powerful in what follows, but there is no getting around the fact that it is quite horrible to look at. It is thus useful to represent it graphically, and a nice notation has been developed in ref. [67]. Consider, for example, the case $m = m' = 2$ (i.e. two incoming and two outgoing particles). Given that we have

$$\langle \mathbf{p}_1', \mathbf{p}_2' | \mathbf{p}_1, \mathbf{p}_2 \rangle \sim \delta^{(3)}(\mathbf{p}_1 - \mathbf{p}_1')\delta^{(3)}(\mathbf{p}_2 - \mathbf{p}_2'),$$

---

[3]Strictly speaking, we mean Poincaré invariant, as we will assume momentum conservation throughout. We have stated Lorentz invariance above, however, as this is the explicit postulate that is usually written in books on this subject.

we can draw this as

$$\langle \mathbf{p}_1', \mathbf{p}_2' | \mathbf{p}_1, \mathbf{p}_2 \rangle \equiv \underline{\qquad}^{\;1}_{\;\;2} \quad , \tag{15}$$

where the horizontal lines indicate that the momenta of the first and second particles remain unchanged. Likewise, we can draw an S-matrix element with two incoming particles as follows:

$$\langle \mathbf{p}_1', \mathbf{p}_2 | \hat{S} | \mathbf{q}_1, \dots \mathbf{q}_n \rangle \equiv \;{}^1_2\!\!\longrightarrow\!\!\bigcirc\!\!S\!\!\longrightarrow\!\!{}^{1}_{2}_{\dots n} \quad . \tag{16}$$

Next, we can introduce *internal lines* in our diagrams, by identifying

$$\underline{\quad\;\; i \quad\;\;} \equiv \int \frac{d^4 p_i}{(2\pi)^3} \delta^{(+)}(p_i^2 - m_i^2). \tag{17}$$

Using this notation, the condition of eq. (14) can be drawn as

$$\underline{\;\;\bigcirc S \;\bigcirc S^\dagger\;} \;+\; \underline{\;\;\bigcirc S \!=\! \bigcirc S^\dagger\;} \;+\; \underline{\;\;\bigcirc S \!\equiv\! \bigcirc S^\dagger\;} \;+\; \dots$$
$$= \underline{\qquad} \;. \tag{18}$$

Here the first line represents the left-hand side of eq. (14), with $m = m' = 2$. The left and right-hand sides of each graph correspond to the two particles with momenta $\{\mathbf{p}_i\}$ and $\{\mathbf{p}_i'\}$ respectively, and the sum over all intermediate states with momenta $\{\mathbf{q}_i\}$ is represented by including an increasing number of internal lines. Each of the latter is associated with an integral as in eq. (17). On the right-hand side of eq. (18), we have two plain lines representing the overlap between two 2-particle states, consistent with eqs. (14, 15). Equation (18) is only one of an infinite number of *unitarity equations*, given that we can draw such a pictorial relation for all possible values of $m$ and $m'$.

We can go further than this, by noting that the fact that we assumed interactions were short-range implies that there is a significant probability that particles will not interact at all. It therefore makes sense to write S-matrix elements in a form that explicitly separates out the contributions where nothing, or something, happens. More specifically, let us take a given initial state $|i\rangle$ and final state $|f\rangle$, and write the corresponding S-matrix element as

$$S_{fi} = \delta_{fi} + i(2\pi)^4 \delta^{(4)}(P_f - P_i) \mathcal{A}_{fi}. \tag{19}$$

Here $\delta_{fi}$ schematically represents the contribution in which the final state is the same as the initial state, which is represented graphically as in eq. (15). For the second term, we have introduced conventional factors of $i$ and $(2\pi)^4$, and extracted an overall delta function which tells us that the momenta of the initial and final states, $P_i$ and $P_f$ respectively, must be equal. This is a consequence of the Poincaré invariance that we assumed earlier. Finally, the quantity $\mathcal{A}_{fi}$ (defined for given initial and final states) encodes the contribution in which there is a genuine interaction between the incoming and outgoing particles, and is known as the *scattering amplitude*. Similar to the graphical notation introduced for the S-matrix above, we may draw eq. (19) as [67]

$$\underline{\;\;\bigcirc S\;} \;=\; \underline{\qquad} \;+\; \underline{\;\;\bigcirc +\;} \quad , \tag{20}$$

where the notation in the second term represents everything in the second term on the right-hand side of eq. (19). By taking the complex conjugate of eq. (19), we find

$$S_{fi}^\dagger = \delta_{fi} - i(2\pi)^4 \delta^{(4)}(P_f - P_i) \mathcal{A}_{fi}^\dagger, \tag{21}$$

which, similarly to the above, we may draw as

$$\text{—}\bigcirc{s^\dagger}\text{—} = \text{———} - \text{—}\bigcirc{\text{–}}\text{—} . \tag{22}$$

The pictures are known as *bubble diagrams* in the literature and old textbooks [39, 40], and we follow the notation of ref. [67] in letting the plus and minus blobs denote the scattering amplitude and its complex conjugate respectively, together with the momentum-conserving delta function and factor of $i(2\pi)^4$.

Equation (20) shows that the S-matrix for $2 \to 2$ scattering decomposes into a trivial term containing two disconnected lines, and a contribution in which the incoming and outgoing particles interact. We may consider a similar concept for any number of incoming and outgoing particles $m$ and $m'$, where the general idea is that one must sum over all possible subsets of interacting particles in the initial and final states. Examples include the following:

$$\text{—}\bigcirc{s}\text{—} = \text{———} + \sum \text{—}\bigcirc{+}\text{—} + \text{—}\bigcirc{+}\text{—} \tag{23}$$

for $m = m' = 3$. Here, the plus bubble notation denotes an amplitude for the interaction of the relevant number of particles entering or leaving the bubble, and the sum in the second term on the right-hand side is over all possible combinations of particles, and thus all interacting subsets. A more non-trivial example for $m = m' = 4$ is

$$\text{—}\bigcirc{s}\text{—} = \text{———} + \sum \text{—}\bigcirc{+}\text{—} + \sum \text{—}\bigcirc{+}\text{—}$$
$$+ \sum \text{—}\bigcirc{+}\bigcirc{+}\text{—} + \text{—}\bigcirc{+}\text{—} , \tag{24}$$

and clearly demonstrates how the combinatorial complexity starts to increase with increasing particle number. The above diagrams show the connectedness structure of a given S-matrix element, and illustrate an important property known as *cluster decomposition*, namely that widely-separated particles should interact independently of each other. This is clearly related to the idea that experiments on opposite sides of the universe should not influence each other, although is not as strong a requirement as strict locality (i.e. the idea that objects can only influence things *immediately* next to them). Nevertheless, cluster decomposition is also a consequence of our assuming that interactions were suitably short-range above. Going beyond S-matrix theory, cluster decomposition plays an important role in arguing that quantum field theory is the unique result of unifying special relativity and quantum mechanics, as stressed in some QFT textbooks [66, 68] [4]. Independently of fields, the combination of unitarity, Lorentz invariance and cluster decomposition has important consequences for the structure of the S-matrix, as we will see. Before moving on, however, we should make explicit another assumption that is often left implicit in the literature on this subject, namely that we are assuming that all of our particle states are massive. This is clearly at odds with some of the physical theories underlying the reality that we appear to live in: electromagnetism and gravity, for example, are apparently carried by massless particles. The assumption is valid, however, for the strong interactions studied in the 1960s, which are presently understood to be described by QCD. The latter theory indeed has a so-called *mass gap*, such that the lowest energy states which carry the interaction at sufficiently long distance are (massive) mesons. We will return to this point in section 4.

---

[4]See ref. [69] for another QFT book with a useful discussion of the implications of unitarity.

## 2.3 Analyticity

Lorentz invariance implies that the scattering amplitude $\mathcal{A}$ (for a given initial and final state) is a function of Lorentz scalars. Given the set of particle momenta $\{p_i\}$ (which may be in- or outgoing) and corresponding masses $m_i$, this means that the amplitude can only depend on the set of invariant variables $\{p_i \cdot p_j, m_i^2\}$. Furthermore, it is a complex function, given that S-matrix elements are in general complex. The physical reason for this is straightforward: particles have wave-like properties in quantum mechanics, and thus any mathematical object that describes their scattering must be able to keep track of phase information. This is precisely what complex numbers are for!

So far, then, we see that the scattering amplitude $\mathcal{A}$ is a complex function of apparently real variables: dot products of 4-momenta (which may be positive or negative), and masses. However, the unitarity and cluster decomposition properties that we introduced in the previous section turn out to imply the following: *scattering amplitudes are the real-boundary values of analytic functions*. This is a somewhat cryptic phrase, so let's decode it carefully. Let us first consider the squared centre of mass energy, which is conventionally denoted by $s$:

$$s = \left( \sum_i p_i \right)^2, \tag{25}$$

where the sum is over all incoming particles $i$. Ordinarily, $s$ is a real number, but let us now imagine extending it to be complex. We can then extend the function $\mathcal{A}(s)$ into the entire complex plane [5], such that its value on the real axis $\text{Im}(s) = 0$ gives the physical amplitude we are seeking. What type of function might this be? We may think of insisting that this function be bounded (i.e. not infinite) and differentiable (i.e. *holomorphic*) in the whole complex plane. However, *Liouville's theorem* of complex analysis tells us that any such function can only be a constant, which clearly does not lead to any interesting scattering! Instead, we conclude that the complex function $\mathcal{A}(s)$ must have singularities, leading to *poles* and *branch cuts* in the complex $s$ plane. Away from these singularities, though, it will be analytic (i.e. expandable in a power series). An example singularity structure is shown in figure 2, and reveals the presence of poles and cuts on the real axis itself. The branch cut is needed because crossing the real axis will be associated with a discontinuity, so that $\mathcal{A}(s)$ becomes multi-valued. The cut reminds us of this, and we must then specify which side of the cut corresponds to the physical amplitude. This turns out to be the upper half plane of $s \in \mathbb{C}$, so that the physical amplitude is the value of $\mathcal{A}(s)$ on the real axis, as approached from above. This is the "real-boundary value" referred to in the above sentence.

Where do the singularities come from? In other words, is there a *physical* explanation for the *mathematical* properties of our particular complex function $\mathcal{A}(s)$? Indeed there is, and we can establish the following two relations:

(a) Cuts arise from *multiparticle thresholds*. That is, if it is only kinematically allowed to make $N$ final states particles above a certain (squared) centre of mass energy $s_N$, we call $\sqrt{s_N}$ the *threshold energy*, and there is a branch point of $\mathcal{A}(s)$ at $s_N$.

(b) Poles are associated with *(bound) single particle states*. If there is a single particle state in the theory - which may be a fundamental or composite particle of mass $M$ - then $\mathcal{A}(s)$ has a pole at $s = M^2$.

We can now see why we expect to find poles and cuts on the real axis e.g. we certainly encounter particle thresholds for physical values of the centre of mass energy, corresponding

---

[5]In general, the amplitude will be a function of more invariants than just $s$. The present discussion then generalises.

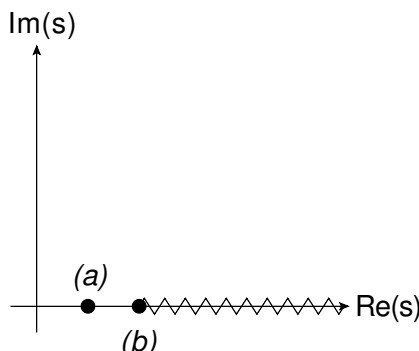

Figure 2: Example singularity structure of the complex scattering amplitude $\mathcal{A}(s)$ in the complex $s$ plane, showing (a) a pole; (b) a branch point.

to $\text{Im}(s) = 0$! By property (a), this will lead to a cut on the real axis, as shown in figure 2. However, neither (a) or (b) is obvious, so let us try to sketch how one might go about proving them.

To derive (a), let us assume we know all possible (multi-)particle states in our theory, and let us take a value of $\sqrt{s}$ such that we can only have two particles in the final state (i.e. we are below the three particle threshold). We may then simplify the unitarity equation of eq. (18) to simply read

$$ \underline{\quad} s \underline{\quad} s^\dagger \underline{\quad} = \underline{\quad\quad} , \tag{26}$$

given that intermediate states with three or more particles (all of which must be on-shell) are no longer allowed to contribute. We can then implement our cluster decomposition property, by replacing each S-matrix element with the amplitude notation of eqs. (20, 22):

$$ \left( \underline{\quad} + \underline{\quad} + \underline{\quad} \right) \left( \underline{\quad} - \underline{\quad} - \underline{\quad} \right) = \underline{\quad} . \tag{27}$$

Remembering what all the symbols mean, we can multiply this out and simplify to get

$$ \underline{\quad} + \underline{\quad} - \underline{\quad} - \underline{\quad} = \underline{\quad} + \underline{\quad} - \underline{\quad} , \tag{28}$$

which, translated back into algebra, means the following:

$$ i(2\pi)^4 \delta^{(4)}(p_1' + p_2' - p_1 - p_2)\big(\mathcal{A} - \mathcal{A}^\dagger\big) \tag{29}$$

$$ = \int \frac{d^4 q_1}{(2\pi)^3} \delta^{(+)}(q_1^2 - m_1^2) i(2\pi)^4 \delta^{(4)}(p_1 + p_2 - q_1 - q_2)\mathcal{A} $$

$$ \times \int \frac{d^4 q_2}{(2\pi)^3} \delta^{(+)}(q_2^2 - m_2^2) i(2\pi)^4 \delta^{(4)}(q_1 + q_2 - p_1' - p_2')(-\mathcal{A}^\dagger). \tag{30}$$

Here we have chosen $\{p_i\}$ and $\{p_i'\}$ to be the incoming and outgoing momenta respectively, and $\{q_i\}$ to be the momenta of the intermediate particles. On the right-hand side, we must include the integrals over the intermediate particle momenta (with the on-shell condition included), and the four-dimensional delta functions enforcing momentum conservation for each sequential part of the diagram: from the initial state to the intermediate state, and from the intermediate state to the final state. Every four-dimensional delta function is accompanied by a factor of $i(2\pi)^4$ according to the conventions outlined above. Finally, each plus or minus

bubble corresponds to a factor of $\mathcal{A}$ or $\mathcal{A}^\dagger$ respectively. We can clearly simplify eq. (30) considerably. For example, we can rearrange the numerical constants, and also combine the two four-dimensional delta functions on the right-hand side, by using up one of the $q_i$ integrals. Following e.g. ref [40], we can then choose to rewrite eq. (30) as

$$i(2\pi)^4\delta^{(4)}(p_1' + p_2' - p_1 - p_2)\big(\mathcal{A} - \mathcal{A}^\dagger\big) = i(2\pi)^4\delta^{(4)}(p_1' + p_2' - p_1 - p_2)$$
$$\times i\int \frac{d^4k}{(2\pi)^4}[2\pi i\delta^{(+)}(k^2 - m^2)][2\pi i\delta^{(+)}(k'^2 - m'^2)]\mathcal{A}\mathcal{A}^\dagger, \tag{31}$$

where we have carried out the $q_2$ integral and defined $k = q_1$, $k' = p_1 + p_2 - k$, so that $k$ and $k'$ are now the 4-momenta in the intermediate state, with masses $m$ and $m'$ respectively. Finally, we may cancel out a common factor to give

$$\mathcal{A} - \mathcal{A}^\dagger = i\int \frac{d^4k}{(2\pi)^4}[2\pi i\delta^{(+)}(k^2 - m^2)][2\pi i\delta^{(+)}(k'^2 - m'^2)]\mathcal{A}\mathcal{A}^\dagger. \tag{32}$$

This has a pleasing form, and indeed one can show that one reproduces eq. (32) by adopting a new set of rules for our plus and minus bubble graphs, which tell us how to translate diagrams featuring multiple bubbles:

(i) Each loop in a plus / minus bubble graph carries a momentum integration

$$i\int \frac{d^4k}{(2\pi)^4}$$

over a *loop momentum k*.

(ii) Each internal line of a graph with momentum $k$ gives a factor

$$2\pi i\delta^{(+)}(k^2 - m^2).$$

(iii) Each plus (minus) bubble carries a factor $\mathcal{A}$ $(-\mathcal{A}^\dagger)$, as we already defined above.

It is relatively straightforward to check that these rules apply for any general bubble diagram containing plus and minus bubbles, and any number of loops. They follow as a direct consequence of the rules for S-matrix diagrams, and the definition of the plus and minus bubbles (see e.g. ref. [40] for a full derivation, or the original work of ref. [67]). Returning to the present case, we see that the right-hand side of eq. (32) is pure imaginary if we are above the two particle threshold (as we have indeed assumed): the factors of $i$ inside the integrand cancel each other out to give $-1$, and the amplitude combined with its complex conjugate will be a positive real number. Another way to see this is to note that the left-hand side of eq. (32) is

$$\mathcal{A} - \mathcal{A}^\dagger = 2i\text{Im}(\mathcal{A}),$$

so that eq. (32) implies

$$2\text{Im}(\mathcal{A}) = \int \frac{d^4k}{(2\pi)^4}[2\pi i\delta^{(+)}(k^2 - m^2)][2\pi i\delta^{(+)}(k'^2 - m'^2)]|\mathcal{A}|^2 \neq 0, \tag{33}$$

where the right-hand side is *real*. We thus conclude that the amplitude $\mathcal{A}$ has a non-zero imaginary part above the two-particle threshold. Below the threshold, the right-hand side of eq. (33) is replaced simply with zero, as it currently assumes that intermediate states with two exchanged particles are kinematically allowed. Hence, $\mathcal{A} \in \mathbb{R}$ on the real axis of the complex

$s$ plane, for values of $\sqrt{s}$ which are below the two-particle threshold [6]. This information is useful, as it allows to almost immediately conclude that not only does $\mathcal{A}$ have an imaginary part for values of $\sqrt{s}$ above the threshold, but it must also be discontinuous across the real $s$ axis. To see this, note that the *Schwarz reflection principle* of complex analysis states that any function $f(s)$ which is real on some part of the real $s$ axis satisfies

$$f^*(s) = f(s^*). \tag{34}$$

In words: the function evaluated with the complex conjugate of its argument, is the same as the complex conjugate of the function of the original argument [7]. Let us now consider our amplitude $\mathcal{A}(s)$ evaluated off the real axis, by shifting its argument into the upper half plane i.e. we will look at $\mathcal{A}(s + i\epsilon)$, $s, \epsilon \in \mathbb{R}$, with $\epsilon > 0$. Equation (33) combined with the Schwarz reflection principle of eq. (34) then implies

$$\text{Im}[\mathcal{A}(s + i\epsilon)] \propto \mathcal{A}(s + i\epsilon) - \mathcal{A}(s - i\epsilon) \neq 0.$$

If we now take $\epsilon \to 0^+$, we see that $\mathcal{A}$ must indeed have a discontinuity across the real $s$ axis, which justifies property (a) above: the function $\mathcal{A}(s)$ has a cut in the complex $s$ plane, associated with the 2-particle threshold. The branch point will be at the value of $s$ such that $\sqrt{s}$ is the centre of mass energy at which two particles can be produced at rest in the final state. Some further comments are in order:

- Here we considered only the two-particle threshold. However, similar arguments can be used to show that there will also be cuts associated with *any $m$-particle threshold*, $m \geq 3$. Alternatively, we can simply take a single branch cut to the right along the real axis, stemming from the two-particle branch point, as this will cover all the other cuts.

- Above we essentially defined the *physical* amplitude as a limit

$$\mathcal{A}(s) = \lim_{\epsilon \to 0^+} \mathcal{A}(s + i\epsilon), \quad s \in \mathbb{R}.$$

  This amounts to saying that the physical amplitude is given by approaching the real axis of $s$ from the upper half plane, and you may be wondering why we did not choose the lower half plane. The answer is that the choice we have made can be (sort of) justified on the grounds of causality, by looking at the scattering of general wavepackets [39,40]. However, such arguments have never been made truly rigorous (although see ref. [70] for an admirable attempt).

- It is worthwhile noting that the prescription used here to define the physical amplitude turns out to agree with the consequences of the Feynman $i\epsilon$ prescription in quantum field theory. This provides some justification for our choice of the upper-half plane, but of course we are trying to construct arguments that are *independent* of any particular theory.

- You might be wondering if it is mysterious that, if we are working in QFT or in pure S-matrix theory, we always need some oddly pedantic prescription involving esoteric factors of $i\epsilon$. It is not mysterious at all - as soon as we surmise that the amplitude must be a complex function in general, and that it must also have discontinuities and therefore potentially be multi-valued, we are *obliged* to state how the physical value of the amplitude is to be obtained, which is bound to end up looking vaguely similar in different approaches.

---

[6]There may still be poles for some values of (real) $s$ below the threshold. However, these are by definition localised at single points, so that there is at least some non-zero region of the real $s$ axis for which $\mathcal{A}$ is itself completely real.

[7]Depending on your disposition, this might be one of those rare cases in which the formula is much clearer than the accompanying words!

Having sketched how to derive property (a) above, let us now turn to property (b), whose proof is perhaps more difficult than you might think it should be, but also rather fun. We start by stating that one can carry out a similar exercise to the derivation of eq. (28) using the unitarity equation for $3 \rightarrow 3$ scattering, where we assume the centre of mass energy is below the four-particle threshold. Combining this with the cluster decomposition of eq. (23), one can derive the following [8]:

$$\text{(diagram)} \tag{35}$$

There are many contributions on the right-hand side, but we will only need to focus on the last one i.e. the sum of terms on the third line. Every term in this sum has a nice physical interpretation: there is a $2 \rightarrow 2$ scattering process, one of whose particles travels some distance and then interacts with another initial particle, in a second $2 \rightarrow 2$ scattering process. We have labelled the 4-momentum of the exchanged particle by $q$ in eq. (35). Using our rules for such diagrams, this contribution has the form

$$\mathcal{A}_{2 \rightarrow 2} \, [2\pi i \delta^{(+)}(q^2 - m^2)](-\mathcal{A}_{2 \rightarrow 2}^{\dagger}), \tag{36}$$

where $\mathcal{A}_{2 \rightarrow 2}$ is the amplitude for $2 \rightarrow 2$ scattering, as the notation suggests. The contribution in the third line of eq. (35) is thus singular for $q^2 = m^2$, and in fact this singularity has a nice physical interpretation: due to the short-range interactions, it is infinitely more likely that the $3 \rightarrow 3$ scattering process occurs via two successive $2 \rightarrow 2$ scatterings, than any other possibility. One can argue on very general grounds [67] that the only way eq. (35) can be satisfied is if the $3 \rightarrow 3$ scattering amplitude (i.e. the first term on the left-hand side of eq. (35)) *also* has a contribution that is singular at $q^2 = m^2$, and which we may draw as

$$\text{(diagram)} \xrightarrow{q^2 \rightarrow m^2} \text{(diagram)}, \tag{37}$$

where we have introduced a dressed internal line factor

$$\text{(diagram)} \;\; \equiv \;\; D^{(+)}(q^2, m^2), \tag{38}$$

and our task is to find the function $D^{(+)}(q^2, m^2)$. This will then show us what kind of singularity an amplitude contains when a single on-shell particle is exchanged. Given eq. (38), we may also write

$$\text{(diagram)} \;\; \equiv \;\; D^{(-)}(q^2, m^2) = [D^{(+)}]^*. \tag{39}$$

That is, as for amplitudes, we use a minus bubble to represent the complex conjugate of the quantity associated with a plus bubble. Armed with these definitions, we may consider eq. (35) in the limit $q^2 \rightarrow m^2$, keeping only those terms which are singular. The result is

$$\text{(diagram)} \tag{40}$$

---

[8]Deriving eq. (35) is an excellent exercise to see if you understand the bubble graph algebra. The full derivation can be found in ref. [40].

To see where this comes from, note that on the left-hand side of eq. (35), each $3 \to 3$ scattering amplitude will be dominated, by its singular piece as $q^2 \to m^2$, which by definition is given by eq. (37) and its complex conjugate. On the right-hand side of eq. (35), the only singular terms (other than in the last line) come from taking the limit of $q^2 \to m^2$ in any $3 \to 3$ scattering amplitudes that appear. There are two of these in the final term in the first line, which give rise to the first two terms on the right-hand side of eq. (40). If we are focusing on the exchange of a single type of particle, only one term in the sum on the last line of eq. (35) contributes i.e. the term in which the particle of interest is exchanged.

We can now simplify eq. (40) as follows. First, we may use the $2 \to 2$ unitarity equation (eq. (28)) on the lower line of the first term on the right-hand side of eq. (40), so that this becomes

$$
\begin{array}{ccc}
\text{(diagram)} & = & \text{(diagram)} - \text{(diagram)}
\end{array} \qquad (41)
$$

Likewise, one has

$$
\begin{array}{ccc}
\text{(diagram)} & = & \text{(diagram)} - \text{(diagram)}
\end{array} \qquad (42)
$$

Substituting eqs. (41) and (42) into eq. (40), we find the simpler relation

$$
\begin{array}{ccc}
\text{(diagram)} - \text{(diagram)} & = & \text{(diagram)}
\end{array} \qquad (43)
$$

which when translated back into algebra yields

$$
\mathcal{A}_{2\to2}\, D^{(+)}(q^2, m^2)\, \mathcal{A}^\dagger_{2\to2} - \mathcal{A}_{2\to2}\, D^{(-)}(q^2, m^2)\, \mathcal{A}^\dagger_{2\to2} = \mathcal{A}_{2\to2}\, [2\pi i \delta^{(+)}(q^2 - m^2)]\, \mathcal{A}^\dagger_{2\to2}.
$$

This shows why the above simplifications have been useful: we now have the *same* factors of a $2 \to 2$ amplitude and / or its complex conjugate in each term on the left- and right-hand sides. We can thus cancel them, to get

$$
D^{(+)}(q^2, m^2) - D^{(-)}(q^2, m^2) = 2\pi i \delta^{(+)}(q^2 - m^2). \qquad (44)
$$

This has a unique solution that is regular in the upper half plane of $q^2$:

$$
D^{(+)} = -\frac{1}{q^2 - m^2 + i\epsilon}. \qquad (45)
$$

Here the $i\epsilon$ ensures that we are indeed in the upper half plane, as was required from causality. Drawing everything together, we have found that the $3 \to 3$ scattering amplitude contains a *pole* at $q^2 = m^2$, where this corresponds to the exchange of a single particle with 4-momentum $q$ and mass $m$. At this pole, the amplitude *factorises* into scattering amplitudes on either side of the exchanged particle, times a function (that of eq. (45)) describing the exchange. Some further comments:

- Here the singularity is in $q^2$ (the exchanged momentum) which, by momentum conservation, will be equal to the invariant mass of all 4-momenta on one side of the exchanged particle. The above argument generalises, for any number of momenta on either side of the exchange, provided we use the corresponding multiparticle unitarity equations.

- The exchanged particle may be fundamental or composite. All that matters in the above argument is that it is some one-particle state of the theory.

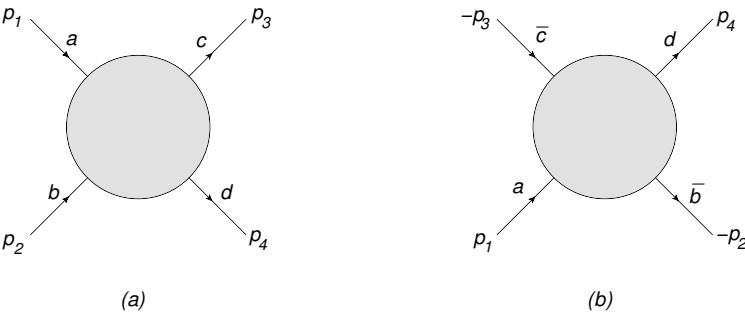

Figure 3: (a) A $2 \to 2$ scattering process, where 4-momenta $\{p_i\}$ and particle species $\{a, b, c, d\}$ are labelled; (b) crossed process, in which (anti-)particle momenta have been exchanged.

- The singularity structure we have derived is not a consequence of unitarity alone, but also of cluster decomposition: we relied crucially on the decomposition of the S-matrix into multiple connected pieces.

- The results above agree with quantum field theory, as they must do given that we are being fully general here. Equation (45) clearly matches the propagator of a scalar particle, if we identify the $i\epsilon$ required to ensure we are in the upper half plane of $q^2$ with the Feynman $i\epsilon$ prescription used to implement causality in QFT. However, the factorisation of an amplitude into smaller amplitudes, times a divergent propagator-like function, also works if the exchanged state is described by a composite operator composed of many fields (e.g. a complicated bound state): see e.g. section 10.2 of ref. [68], or section 24.3 of ref. [66].

In summary, we have established that the S-matrix must be a complex function of Lorentz scalars, where the latter are themselves considered as complex variables. The amplitude will then have poles and cuts associated with bound states and multiparticle thresholds respectively. Why, though, is this useful? Firstly, we will see some explicit examples of amplitudes having poles in the squared centre of mass energy $s$ later on. We will then be able to directly interpret these in terms of bound states! Secondly, knowing the singularity of the structure of the amplitude $\mathcal{A}$ is a crucial preqrequisite for being able to analytically continue this function. We will need to do this later, when describing high energy behaviour. Furthermore, the ability to continue a function to different *regions* of its kinematic variables allows us to conclude that the amplitudes for different scattering processes are in fact related. We explore this in the following section.

## 2.4 Physical regions and crossing

We have seen that amplitudes depend on Lorentz invariants, regarded as complex variables. However, not all regions of the complex plane of each Lorentz invariant are kinematically accessible. To illustrate this, let us consider the $2 \to 2$ scattering process

$$a(p_1) + b(p_2) \to c(p_3) + d(p_4) \tag{46}$$

depicted in figure 3. For simplicity, we will consider a case in which all particles have equal mass, so that

$$p_i^2 = m^2 \quad \forall i.$$

For any $2 \to 2$ process, it is conventional to define the so-called *Mandelstam invariants*

$$s = (p_1 + p_2)^2, \quad t = (p_1 - p_3)^2, \quad u = (p_1 - p_4)^2. \tag{47}$$

We have seen the first of these already: it represents the squared total energy in the centre of mass frame. Furthermore, the second invariant $t$ has a nice interpretation as the square of the total 4-momentum exchanged between the incoming particles. The third invariant $u$ is not independent from the others. Using the momentum conservation condition

$$p_1 + p_2 = p_3 + p_4, \tag{48}$$

one may derive the fact that

$$s + t + u = \sum_{i=1}^{4} m_i^2 = 4m^2. \tag{49}$$

Let us now see what the region of the $(s, t, u)$ space is that corresponds to physically allowed scattering. We can choose any frame to examine this, given that each parameter is Lorentz invariant. Let us then choose the centre of mass frame, in which one may parametrise the incoming and outgoing momenta according to

$$p_1 = (E, \mathbf{p}), \quad p_2 = (E, -\mathbf{p}), \quad E^2 - \mathbf{p}^2 = m^2, \tag{50}$$

and

$$p_3 = (E', \mathbf{p}'), \quad p_4 = (E', -\mathbf{p}'), \quad (E')^2 - (\mathbf{p}')^2 = m^2, \tag{51}$$

respectively. Energy conservation then implies that $E = E'$. Furthermore, there will be some angle $\theta$ between the 3-momenta $\mathbf{p}$ and $\mathbf{p}'$ in general, so that we have

$$\begin{aligned}
s &= 4E^2 = 4(|\mathbf{p}|^2 + m^2), \\
t &= -2|\mathbf{p}|^2(1 - \cos\theta), \\
u &= -2|\mathbf{p}|^2(1 + \cos\theta).
\end{aligned} \tag{52}$$

Physical scattering corresponds to $E \geq 0$ and $|\cos\theta| \leq 1$, thus to the region

$$s \geq 4m^2, \quad t \leq 0, \quad u \leq 0. \tag{53}$$

This is a highly restricted region of the total $(s, t, u)$ space. However, other regions of this space can also be given a physical meaning. Imagine, for example, rotating the diagram of figure 3(a) to get the diagram of figure 3(b). One can easily see from rotating the diagram that $p_1$ and $p_4$ remain incoming and outgoing momenta respectively. However, $p_3$ switches from the final state to the initial state. If we want the 4-momentum to be flowing *in* to the scattering process, we must reverse the sign of $p_3$. Likewise, $p_2$ moves to the final state, and again must be reversed in sign if we want the momentum to be flowing *out* of the process. In figure 3(b), we have also allowed for the particle species to change in general, if a particle moves from the initial to the final state or vice versa i.e. we have replaced $(b, c) \to (\bar{b}, \bar{c})$. To see how they are related, consider that there is some additive quantum number possessed by all the particles e.g. a conserved charge $Q$ of some kind. The existence of the scattering process in figure 3(a) then implies

$$Q_a + Q_b = Q_c + Q_d. \tag{54}$$

Similarly, the process of figure 3(b) implies

$$Q_a + Q_{\bar{c}} = Q_{\bar{b}} + Q_d, \tag{55}$$

and comparison of eqs. (54) and (55) implies [9]

$$Q_{\bar{b}} = -Q_b, \quad Q_{\bar{c}} = -Q_c. \tag{56}$$

---

[9]Equation (56) is actually a stronger condition than that directly implied by comparing eq. (54) with eq. (55). However, one can consider swapping the particles from the final to the initial state (or vice versa) one by one, in which case eq. (56) does indeed follow.

This argument will work for any such quantum number, and we rapidly conclude that $\bar{b}$ and $\bar{c}$ must be the *antiparticles* of $b$ and $c$. From the figure, we see that the new process corresponds to the old one (i.e. it is the same amplitude, just rotated) with the replacements $b \to \bar{c}$, $c \to \bar{b}$, and $s \leftrightarrow t$. It will thus have a physical region

$$t \geq 4m^2, \quad s \leq 0, \quad u \leq 0, \tag{57}$$

and we expect

$$\mathcal{A}_{ab \to cd}(s, t, u) = \mathcal{A}_{a\bar{c} \to \bar{b}d}(t, s, u). \tag{58}$$

Clearly this is a different region of $(s, t, u)$ space than that of eq. (53), as might be expected given that our two different physical regions are meant to correspond to physically distinct scattering processes. Switching between processes that correspond to different regions of the same amplitude function is called *crossing*, and the fact that the amplitudes of eq. (58) are equal is called *crossing symmetry*. However, we have to be very careful about whether this is really true, despite the fact that the above arguments might look very convincing! We have in fact made an assumption in eq. (58), namely that one can safely analytically continue the amplitude in one physical region into another (n.b. this happens when we send $(s, t) \to (t, s)$). As we send $s$ or $t$ from one region of their respective complex planes to another, we might encounter singularities, that disrupt the simple relationship of eq. (58), in that we might not be able to interpret the function we get in the new region as a scattering amplitude. However, provided that the only singularities we encounter are the poles and cuts we already found, and that we know amplitudes are meant to have, then everything should be OK. In asserting crossing symmetry, we thus assume that the only singularities in the entire complex plane of any Mandelstam invariant are the poles and cuts associated with bound states and thresholds. This is sometimes called *maximal analyticity of the first kind* [39], which is a rather silly and fancy name, but at least makes clear that an assumption is being made [10]. Once we have crossing symmetry, however, we can surmise the presence of cuts and poles in the complex $t$ and $u$ planes, which are obtained from their counterparts in the $s$ plane by the appropriate crossing relationships.

Above, we interchanged $s$ and $t$ and obtained a different process. We could also have performed a different crossing, and obtained

$$\mathcal{A}_{ab \to cd}(s, t, u) = \mathcal{A}_{\bar{d}b \to c\bar{a}}(u, t, s). \tag{59}$$

Here $u$ and $s$ have been interchanged, so we conclude from eq. (53) that we have produced a new physical region, defined by

$$u \geq 4m^2, \quad t \leq 0, \quad s \leq 0. \tag{60}$$

One way to draw all these possible physical regions is on a so-called *Mandelstam diagram*, which has $(s, t, u)$ axes mutually aligned at $60°$ to one another. This is shown, for our example of equal masses, in figure 4. The three physical regions we have identified in eqs. (53, 57, 60) are known as the "$s$-channel", "$t$-channel" and "$u$-channel" respectively, where the parameters $(s, t, u)$ increase in the directions shown. The $s$-channel starts at $s = 4m^2$, as we found above, and we illustrate this on the figure. Such figures are a tad obfuscating at first glance, particularly given that they have long since fallen out of fashion. However, they do provide a nice way to visualise all physical regions at once. Note that the boundaries of the physical regions in figure 4 are particularly nice (i.e. straight lines), due to our choice of equal masses. Things are more complicated for general particles / masses, as you may like to investigate yourself! In such cases, there will still be three physical regions, and we can still talk about analytic continuation from one channel to another.

---

[10]There is actually an even more subtle assumption being made, namely that swapping $s$ and $t$ takes us from the physical Riemann sheet of one complex amplitude to the physical sheet of another. It is best to simply say that crossing symmetry is *assumed* rather than *derived*. Perturbative amplitudes in QFT certainly respect it.

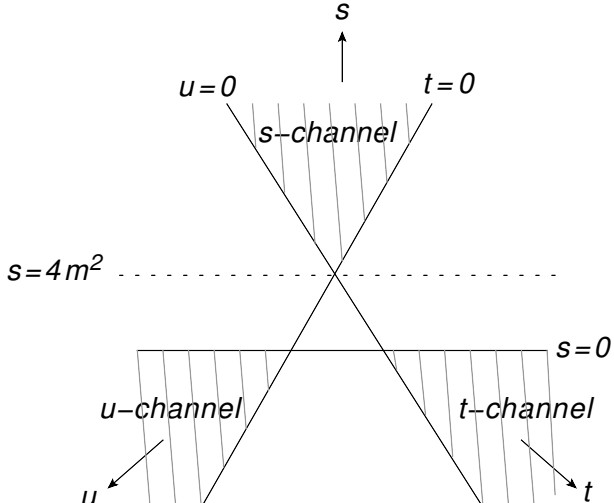

Figure 4: Mandelstam diagram showing the three physical regions of $2 \to 2$ scattering, in the case where all particle masses are equal.

## 3 Regge Theory

### 3.1 The Regge limit

So far, we have studied general structures of amplitudes (e.g. singularities, crossing symmetry) and how these arise from general assumptions (e.g. superposition, Lorentz invariance, unitarity, localised interactions). In this section, we will focus on a particular kinematic region, and see what else we can learn. In particular, we will study the high-energy or *Regge limit* of $2 \to 2$ scattering [11]. Defining invariants as in eq. (47), we can formally define the Regge limit as

$$s \gg -t \gg m^2. \tag{61}$$

Here we are working in the physical $s$-channel region of eq. (53), for which $t$ will be negative. For simplicity, we will continue to assume common particle masses $m$, but this will not matter too much for what follows. Physically, the above limit corresponds to the centre of mass energy being much larger than the momentum exchanged by the scattering particles, which corresponds to highly forward scattering, in which the colliding particles barely glance off each other. Furthermore, the requirement of high energy does not itself place a restriction on the relative ordering of the (squared) momentum transfer $|t|$, and the squared particle masses $m^2$. We have chosen a particular ordering above, but you might sometimes see the Regge limit being defined with the alternative choice:

$$s \gg m^2 \gg -t.$$

Which choice one makes ultimately depends on whether one wants to keep track of mass information, including allowing a certain mass to become large. For our present discussion, this will not be important, and we will therefore stick with the limit of eq. (61).

Our reason for studying the high energy limit is that it has a number of applications in non-abelian gauge theories and gravity. In collider physics, for example, present-day experiments such as the LHC (and to some extent the Tevatron and HERA beforehand) probe kinematic

---

[11]The use of the phrase *high energy* for this limit is something of an abuse of terminology, albeit one that is common throughout the literature. We will use the terms *high energy* and *Regge* interchangeably throughout.

regions in which additional contributions arising from the Regge limit become relevant. This may affect the description of the quark and gluon distributions within the proton [54–64], for example, or the radiation profile of multiple jets [43–53, 71–74].

In gravity, the high energy limit is related to scattering above the Planck scale (i.e. *transplanckian scattering*) [65, 75–88], and also to high energy collisions of black holes, which have become extremely topical due to the recent discovery of gravitational waves by LIGO.

We will see that it is possible to say quite a lot about the high energy behaviour of scattering amplitudes, based solely on general principles. The main idea involved is that large $s$ behaviour in the $s$-channel is related to particle exchange in the $t$-channel. To make sense of this statement, let us first note that it is possible to write the amplitude in the physical $s$-channel as a sum of contributions with definite angular momentum in the $t$-channel:

$$\mathcal{A}(s,t) = 16\pi \sum_{l=0}^{\infty} (2l+1) a_l(t) P_l(z_t). \tag{62}$$

Here the overall factor of $16\pi$ is purely conventional, as is the factor of $(2l+1)$ in each term (although it has a simple interpretation as the total number of states with orbital angular momentum $l$). The function $P_l(z_t)$ is a *Legendre polynomial* involving the $t$-channel scattering angle $\theta_t$ via the argument

$$z_t = \cos\theta_t. \tag{63}$$

You may recall that the Legendre polynomials are used in the solution of the hydrogen atom in quantum mechanics, where they arise specifically as the eigenfunctions of orbital angular momentum. Here the principle is similar: whatever the amplitude is, it must be expandable in terms of a complete set of eigenfunctions involving the $t$-channel scattering angle. These eigenfunctions are precisely the Legendre polynomials. The final ingredient of each term in eq. (62) is $a_l(t)$, which tells us "how much" of each Legendre polynomial we have. This will depend upon $t$ in general, given that we have expressed the amplitude in terms of definite angular momentum in the $t$-channel. Equation (62) is called a *partial wave expansion*, and the coefficients $\{a_l(t)\}$ are called *partial wave amplitudes*. You may already be familiar with this from the study of scattering theory in non-relativistic quantum mechanics. If not, we can surmise eq. (62) on very general grounds, as described above.

Equation (62) is a bit unusual, in that we are talking about the amplitude in the physical $s$-channel, but expressed as a $t$-channel partial wave amplitude. This is because we are trying to justify the statement made above, that exchanges in the $t$-channel can be related to definite high energy behaviour in the $s$-channel. To go further, we need the following formula for the $t$-channel scattering angle in terms of the Mandelstam invariants $s$ and $t$:

$$\cos\theta_t = 1 + \frac{2s}{t - 4m^2}. \tag{64}$$

To see where this comes from, note that the scattering angle in the $s$-channel is given implicitly in eq. (52), from which we find

$$|\mathbf{p}|^2 = -\frac{t}{(1-\cos\theta_s)} = \frac{s}{4} - m^2 \quad \Rightarrow \quad \cos\theta_s = 1 + \frac{2t}{s - 4m^2}.$$

The result of eq. (64) then follows from crossing symmetry, and our $s$-channel amplitude becomes

$$\mathcal{A}(s,t) = 16\pi \sum_{l=0}^{\infty} (2l+1) a_l(t) P_l\left(1 + \frac{2s}{t - 4m^2}\right). \tag{65}$$

Now let us take the case where a single particle of definite spin $J$ is exchanged in the $t$-channel, as in figure 5. Note that this is not, strictly speaking, a Feynman diagram, as we are

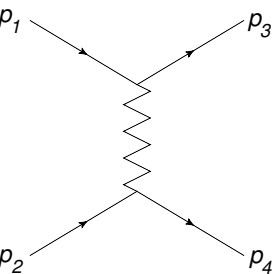

Figure 5: Exchange of a particle of spin $J$ in the $t$-channel of $2 \to 2$ scattering, where we consider the physical scattering (from left to right) in the $s$-channel. Strictly speaking, this is not a Feynman diagram, given that we are not necessarily in a QFT.

not necessarily in a QFT. Even if we were, the exchanged particle may be some complicated non-perturbative bound state, rather than one of the fundamental particles in the theory. As we saw in section 2, amplitudes are singular if a particle is exchanged in a given channel. Thus, the expansion of eq. (65) is dominated by a single term, namely the one corresponding to $l = J$ (i.e. the angular momentum in the $t$-channel must correspond to the exchanged spin):

$$\mathcal{A}(s,t) \to 16\pi(2J+1)a_J(t)P_J\left(1 + \frac{2s}{t-4m^2}\right).$$

We can now take the Regge limit of eq. (61), using the following known asymptotic behaviour of Legendre polynomials:

$$\lim_{z\to\infty} P_l(z) \sim z^l. \tag{66}$$

We conclude that

$$\mathcal{A}(s,t) \sim \left(\frac{s}{t}\right)^J. \tag{67}$$

In words: *power-like growth of amplitudes with squared centre of mass energy s is associated with exchange of (bound) particle states in the t-channel.* Our wording stresses that the exchanged particles may be fundamental or composite.

The above argument is, of course, a simplification, given that it relied on the exchange of only one particle state. In practice, many particles may be exchanged in the $t$-channel, so that multiple partial waves contribute. One must then consider the full series of eq. (65), and here there is a problem: if we naïvely try to carry out the sum of eq. (65) in the physical $s$-channel region, it does not converge! The origin of this problem is that the $t$-channel partial wave expansion, strictly speaking, only applies in the physical region of $t$-channel scattering, namely that of eq. (57). However, we are choosing to apply it in the physical $s$-channel region of eq. (53), with nary a thought as to whether or not it has been correctly analytically continued. There is a truly ingenious way to get around this problem that was first proposed by Regge in non-relativistic quantum mechanics [89]. It involves rewriting the series of eq. (65) as a complex integral, that can then be made to give sensible results in the physical $s$-channel region by deforming it. The upshot is that if a theory contains a spectrum of states that can be exchanged in the $t$-channel, then this indeed leads to generic high energy behaviour in the $s$-channel.

## 3.2 Complex angular momentum

Quite obviously, the angular momentum $l$ in eq. (65) must be a non-negative integer. Regge's idea was instead to let this be a complex variable. This is not such a daft idea, given that we have already let Lorentz invariants (formed from linear momenta) become complex. How,

though, do we achieve a complex $l$? The first step is to replace the partial wave amplitudes in eq. (65) as follows:

$$\{a_l(t)\} \rightarrow a(l, t),$$

where on the right-hand side we have a complex function $a(l, t)$ of $l, t \in \mathbb{C}$, whose only requirement is that it reproduces the partial wave amplitudes if $l$ is a non-negative integer:

$$a(l, t) \rightarrow a_l(t), \quad l \in \{0, 1, 2, \ldots\}.$$

Unfortunately, this does not look unique: naïvely, it seems we have the freedom to replace

$$a(l, t) \rightarrow a(l, t) + f(l, t),$$

where

$$f(l, t) = 0, \quad l \in \{0, 1, 2, \ldots\}.$$

If the procedure is not unique, how then can we be confident of getting a definite answer for the $s$-channel scattering amplitude, which certainly ought to be unique? The resolution lies in something called *Carlson's theorem*, which says that *if two different analytic functions do not grow too fast at infinity, they cannot coincide at non-negative integers.* In other words, $a(l, t)$ is *unique*, provided it does not grow too fast. The precise criterion that Carlson requires is that

$$a(l, t) < C e^{\pi |l|}, \quad |l| \rightarrow \infty,$$

where $C$ is an arbitrary constant. Unfortunately, this is not quite satisfied! One can show that the partial wave amplitudes $\{a_l(t)\}$ have contributions that alternate in sign, so that

$$a_l(t) \sim (-1)^l = e^{-i\pi l}, \tag{68}$$

which clearly violates the requirements of Carlson's theorem on the imaginary axis. We can rescue things by separating the odd and even partial waves: each of these has contributions associated with the *same* sign, which will not then behave like eq. (68), and thus we can use Carlson's theorem to guarantee uniqueness of our complexified odd / even partial wave expansions. The conventional way to do this is to use the property of Legendre polynomials

$$P_l(-z) = (-1)^l P_l(z) \tag{69}$$

to define

$$\mathcal{A}^{\pm}(s, t) = 8\pi \sum_{l=0}^{\infty} (2l + 1) a_l^{\pm}(t) [P_l(z_t) \pm P_l(-z_t)]. \tag{70}$$

This defines the even (odd) amplitude $\mathcal{A}^+$ ($\mathcal{A}^-$), where the factor in the square brackets will pick out even (odd) partial waves only for the upper (lower) sign. Furthermore, we have defined the even and odd partial wave amplitudes $a^{\pm}(t)$, such that

$$a_l(t) = \begin{cases} a_l^+(t), & l \text{ even}; \\ a_l^-(t), & l \text{ odd}. \end{cases}$$

We do not have to worry about defining $a_l^+(t)$ for $l$ odd (or $a_l^-(t)$ for $l$ even), given that such terms will be killed off by the combination of Legendre polynomials in eq. (70). The full amplitude is then given by

$$\mathcal{A}(s, t) = \mathcal{A}^+(s, t) + \mathcal{A}^-(s, t). \tag{71}$$

We now have two different partial wave expansions, involving two sets of partial wave amplitudes $\{a_l^+(t)\}$ and $\{a_l^-(t)\}$, each of which separately satisfies the requirements of Carlson's

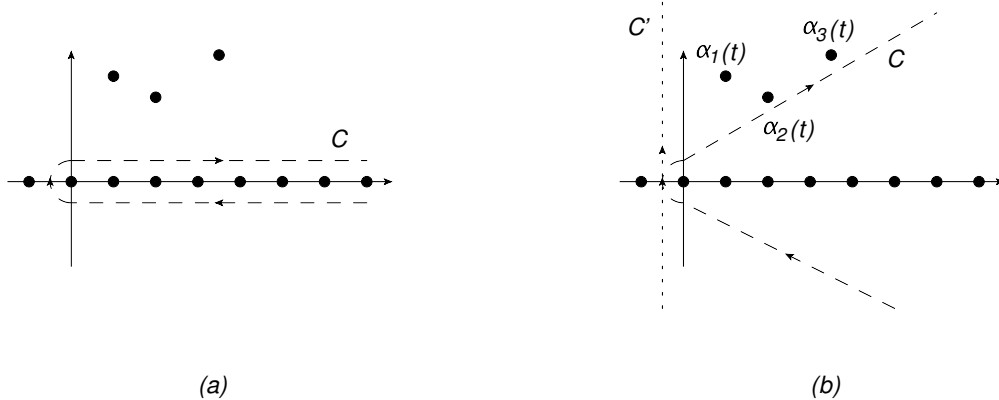

(a)                                        (b)

Figure 6: (a) Pole structure of the integrand in eq. (72), where poles in the real axis arise from zeroes of the sin function; (b) Deformation of the original integration contour $C$ (enclosing the real axis) to make a vertical contour $C'$, which is closed at infinity.

theorem. We can thus define two *unique* functions $a^+(l,t)$ and $a^-(l,t)$ in the complex $l$ plane, such that $a^+(l,t)$ reduces to $a_l^+(t)$ for even $l$, and $a^-(l,t)$ to $a_l^-(t)$ for odd $l$. Next, we can rewrite the expansions of eq. (70) as complex integrals [12]:

$$\mathcal{A}^\pm(s,t) = -4\pi i \int_C dl (2l+1) a^\pm(l,t) \frac{[P_l(-z_t) \pm P_l(z_t)]}{\sin(\pi l)}, \tag{72}$$

where $C$ is a contour that completely encloses the positive real axis, as shown in figure 6(a). To justify this formula, note that the function $\sin(\pi l)$ in the denominator has zeroes at all integer values of $l$. Thus, there is a series of equally spaced poles of the integrand on the real $l$ axis, at all values $l \in \mathbb{Z}$. Each of these is in fact a simple pole, given that one has

$$\sin(\pi l) \to (-1)^n (l-n)\pi + \mathcal{O}[(l-n)^2], \quad n \in \mathbb{Z}.$$

This also allows to find the residue of each pole, using the general result for simple poles

$$\text{Res}_{z=z_0} f(z) = \lim_{z \to z_0} (z-z_0) f(z).$$

Denoting the integrand of eq. (72) by $f(l)$, we then have

$$\text{Res}_{l \to n \in \mathbb{Z}} f(l) = \lim_{l \to n} \frac{(-4\pi i)}{(-1)^n} \frac{(l-n)}{(l-n)\pi} (2l+1) a^\pm(l,t) [P_l(-z_t) \pm P_l(z_t)]$$

$$= -\frac{4\pi i}{(-1)^n \pi} (2n+1) a^\pm(n,t) [P_n(-z_t) \pm P_n(z_t)]. \tag{73}$$

We can then carry out the complex integral of eq. (72) using *Cauchy's theorem*

$$\oint_C f(z) = 2\pi i \sum_i \text{Res}_{z=z_i} f(z), \tag{74}$$

where $\{z_i\}$ denotes the locations of all poles. In our present case this gives

$$\mathcal{A}^\pm(s,t) = 2\pi i \sum_{n=0}^\infty \text{Res}_{l=n} f(l)$$

$$= 8\pi \sum_{n=0}^\infty (2n+1) a_n^\pm(t) [P_n(z_t) \pm P_n(-z_t)], \tag{75}$$

---

[12] In eq. (72), the Legendre polynomials are understood to be replaced by their analytic continuations for complex $l$, for which a known form in terms of hypergeometric functions exists.

where we have used eq. (69), and the fact that $a^{\pm}(l, t)$ is the same as $a_l^{\pm}(t)$ for even or odd $l$ respectively. We have thus indeed shown that the complex integral representation of eq. (72) reproduces the $t$-channel even and odd partial wave expansions of eq. (70). Equation (72) has a name in the literature: it is known as the *Sommerfeld-Watson transform*.

## 3.3 Regge poles

So far, we have made the claim that high energy behaviour in the $s$-channel should be dictated by particle exchange in the $t$-channel. This led us to write the amplitude in the $s$-channel as a $t$-channel partial wave expansion, which however did not converge. Thus, we then rewrote the expansion in terms of two highly complicated complex integrals in the angular momentum plane, where our ultimate aim is to find something that indeed converges in the physical $s$-channel region. It is, of course, not immediately clear why the rewriting inherent in the Sommerfeld-Watson transform is useful. However, the power of having a complex integral representation (eq. (72)) instead of a discrete sum (eq. (70)) is that we can deform the contour of integration to get an $s$-channel representation of the scattering amplitude that has better convergence properties. Put another way, deforming the contour of our Sommerfeld-Watson transform amounts to carefully analytically continuing the amplitude from the physical $t$-channel region to the $s$-channel region, so that we can safely study its properties in the Regge limit of eq. (61).

The original contour $C$ in figure 6(a) encloses the real axis (n.b. it is assumed to be closed at positive infinity on the real $l$ axis). Let us deform it, as in figure 6(b), to a vertical contour $C'$ at $Re(l) = -1/2$, which is then closed to the right at infinity. In doing this, we will end up moving the contour past any other singularities that might be present in the complex plane, and that are not associated with the simple poles along the real axis from the denominator in eq. (72). Instead, they would be associated with the partial wave functions $a^{\pm}(l, t)$ sitting in eq. (72). For now, we will assume that only simple poles are present. Given that they are contained in the partial wave functions, we must include the possibility that the location of these additional poles in the complex angular momentum plane can depend on $t$. We will thus label the positions of the poles by $\{\alpha_i(t)\}$, as exemplified by figure 6(b).

By the usual tricks of complex analysis, moving the contour $C$ past a simple pole means that we pick up its residue. In total, then, deforming the contour $C \to C'$ means that we pick up the residues of all poles $\{\alpha_i(t)\}$ lying between $C$ and $C'$. Assuming the amplitude vanishes at infinity (so that there is no contribution from the contour integral there) eq. (72) becomes, after the contour deformation,

$$
\mathcal{A}^{\pm}(s, t) = 8\pi^2 \sum_i \frac{(2\alpha_i^{\pm}(t) + 1)\beta_i^{\pm}(t)}{\sin[\pi\alpha_i^{\pm}(t)]} \left[ P_{\alpha_i^{\pm}}(-z_t) \pm P_{\alpha_i^{\pm}}(z_t) \right]
$$
$$
- 4\pi i \int_{-1/2-i\infty}^{1/2+i\infty} dl(2l + 1)a^{\pm}(l, t) \frac{[P_l(-z_t) \pm P_l(z_t)]}{\sin(\pi l)}. \tag{76}
$$

Here the first line arises from collecting the residues of all the poles $\{\alpha_i(t)\}$, where $\beta_i^{\pm}(t)$ is a residue factor. We have no way of knowing what the latter is without knowing the partial wave amplitudes (which we are claiming to know nothing about). However, the residue factor can depend only on $t$, which still amounts to non-zero information. The second line in eq. (76) corresponds to the remaining complex integral over the contour $C'$. Whilst it may not seem like it at first glance, we have indeed finally put the $s$-channel amplitude in a form that allows us to safely examine the Regge limit. Furthermore, it simplifies considerably!

Let us first consider the first line of eq. (76), namely the pole contributions. Recalling

eq. (64), we find in the Regge limit of eq. (61) that [13]

$$z_t \to \frac{2s}{t}. \tag{77}$$

Given $t < 0$ in the physical $s$-channel region, we see that $z_t$ is large and negative. We may utilise the known relation for Legendre polynomials [14]

$$P_\alpha(z_t) = e^{-i\pi\alpha} P_\alpha(-z_t) \tag{78}$$

to write

$$P_\alpha(-z_t) \pm P_\alpha(z_t) = [1 \pm e^{-i\pi\alpha}] P_\alpha(-z_t). \tag{79}$$

Then, using the asymptotic behaviour

$$P_\alpha(x) \xrightarrow{x \gg 1} \frac{\Gamma(2\alpha+1)}{\Gamma^2(1+\alpha)} \left(\frac{x}{2}\right)^\alpha, \quad \mathrm{Re}(\alpha) \geq \frac{1}{2}, \tag{80}$$

where $\Gamma(z)$ is the Euler gamma function, the contribution in the first line of eq. (76) becomes

$$8\pi^2 \sum_i \frac{(2\alpha_i^\pm + 1)\beta_i^\pm(t)}{\sin[\pi\alpha_i^\pm(t)]} \left(1 \pm e^{-i\pi\alpha_i^\pm(t)}\right) \frac{\Gamma(1+2\alpha_i^\pm)}{\Gamma^2(1+\alpha_i^\pm)} \left(\frac{s}{-t}\right)^{\alpha_i^\pm}. \tag{81}$$

We may tidy this up further using *Euler's reflection formula* [15]

$$\Gamma(1+z)\Gamma(-z) = -\frac{\pi}{\sin(\pi z)}, \tag{82}$$

so that eq. (81) becomes

$$\sum_i \tilde{\beta}_i^\pm(t) \Gamma[-\alpha_i^\pm(t)] \left(1 \pm e^{-i\pi\alpha_i^\pm(t)}\right) \left(\frac{s}{-t}\right)^{\alpha_i^\pm}, \tag{83}$$

where we have absorbed some factors in the residue factor (hence the tilde). We must also worry about the remaining complex integral in the second line of eq. (76), which is known as the *background integral* in old literature on this subject. However, Mandelstam has shown [90] that this can be made arbitrarily small, at the expense of including extra poles $\alpha_i^\pm(t)$ in the above sum. Thus, the form of the result is unchanged, and our complete result for the $s$-channel physical amplitude in the Regge limit is

$$\mathcal{A}^\pm(s,t) = \sum_i \tilde{\beta}_i^\pm(t) \Gamma[-\alpha_i(t)] \left(1 \pm e^{-i\pi\alpha_i^\pm(t)}\right) \left(\frac{s}{-t}\right)^{\alpha_i^\pm(t)}. \tag{84}$$

We have obtained a remarkable result. Based purely on well-motivated assumptions for how the S-matrix should behave, we have found that the behaviour of the $2 \to 2$ scattering amplitude in the high energy (Regge) limit of eq. (61) consists of a series of power-like terms in $s$, with $t$-dependent prefactors. Each power corresponds to a pole in the complex angular momentum plane, that moves about as a function of $t$. In fact, we can simplify this even further: if all we care about is the *leading* behaviour as $s \to \infty$, we can keep only the term in eq. (84) whose power has the largest real part. In other words, instead of summing over all poles in eq. (84), we can keep only the right-most pole $\alpha(t)$ in the complex $l$ plane. This might occur

---

[13]Equation (77) would be different had we defined the Regge limit such that the momentum transfer was much less than the particle masses.

[14]Equation (78) differs from eq. (69) in that it applies for arbitrary complex $\alpha$.

[15]Equation (82) is usually written with $z \to -z$.

in either the even or the odd amplitude, in which case it will dominate the result for the total amplitude of eq. (71). Let us introduce a factor $\eta = \pm 1$, called the *signature* of the right-most pole, where the upper (lower) sign corresponds to its being in the even (odd) amplitude. Then the total amplitude will be given by

$$\mathcal{A}(s,t) \xrightarrow{s \gg -t} \tilde{\beta}(t)\Gamma[-\alpha(t)]\left(1 + \eta e^{-i\pi\alpha}\right)\left(\frac{s}{-t}\right)^{\alpha(t)}. \tag{85}$$

You might be wondering, given the incredibly convoluted series of mathematical steps that it took to get this result, whether there is any hope of understanding the physics of what is going on. Indeed there is! We saw already above that power-like growth with $s$ in the $s$-channel can be identified with the exchange of single particle states in the $t$-channel. We saw this for the case of a *single* particle being exchanged, but in fact eq. (85) corresponds to infinitely many states being exchanged in the $t$-channel. To see why, note that the Euler gamma function has poles at all non-positive integers $z \in \{0, -1, -2, -3, \ldots\}$. Thus, the factor of $\Gamma[-\alpha(t)]$ in eq. (85) implies that the amplitude $\mathcal{A}(s,t)$ has poles in $t$, whenever $t$ is such that $\alpha(t)$ is a physical angular momentum. But, as we derived in section 2, poles in any Lorentz invariant represent the exchange of bound states! Thus there is an infinite family of bound states that can be exchanged in the $t$-channel, and it is the exchange of this entire family that gives the power-like growth in $s$ in the Regge limit.

We can really think of the entire family of bound states as being contained in the function $\alpha(t)$: this parametrises a curve in the complex angular momentum plane, such that whenever the curve crosses a physical value of Re($l$) (i.e. $l \in \{0, 1, 2 \ldots\}$), the corresponding value of $t$ is the squared mass of a state of spin $l$. Returning to the case where we consider many different simple poles, each function $\alpha_i(t)$ is called a *Regge trajectory*. A commonly-performed exercise in the 1960s was to figure out which known mesons (bound states) lay on the same Regge trajectory, and to use the measured values of their spin and masses to construct the function $\alpha(t)$ for that particular trajectory. For mesons, the functions were found to be linear to a very good approximation, which happens to be exactly the relation one would expect if mesons were described by rotating strings. This is how string theory was born: see chapter 1 of ref. [91] for a nice historical overview.

Above, we only admitted the possibility of single poles in the complex angular momentum plane. The arguments are relatively straightforward to repeat, however, in the presence of other types of singularity. It can be shown [39, 40], for example, that higher-order poles dress the above behaviour according to

$$\mathcal{A}(s,t) \sim \log^n(s) s^{\alpha(t)}, \tag{86}$$

for some integer $n$, and that cuts lead to the behaviour

$$\mathcal{A}(s,t) \sim s^{\alpha_c(t)}(\log s)^{-\gamma(t)}, \tag{87}$$

where $\alpha_c(t)$ is the location of the branch point on the complex $l$ plane, and $\gamma(t)$ is related to the discontinuity across the cut.

In this section, we have shown that generic behaviour arises in the Regge limit of scattering amplitudes, namely power-like growth in the centre of mass energy (up to possible logarithmic corrections). This behaviour is traced to the exchange of particle states in the $t$-channel, and thus will only occur if such states exist in the theory. Note that, in the absence of an underlying theory, we cannot say what the singularities in the complex angular momentum plane will be. One can then "guess" a possible structure of singularities, before fitting the results to data. This approach is still used today for fitting certain observables in strong interaction physics which lack a perturbative description e.g. total cross-section data at the LHC [92, 93].

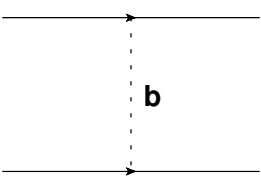

Figure 7: In the Regge limit of $2 \to 2$ scattering, the incoming particles barely glance off each other, thus follow straight-line trajectories separated by an *impact parameter* **b**.

Given the above behaviour is completely generic, it must also be the case that any particular underlying theory - such as a perturbative quantum field theory - must reproduce the power-like growth in $s$ in the high energy limit. In the following section, we will show examples of this in both QCD and gravity, and examine the relationship between them.

## 4 Examples in QCD and gravity

In this section, we will examine the high energy behaviour of scattering amplitudes in two widely studied theories, namely QCD (relevant for collider physics) and gravity (relevant for astrophysics and cosmology). Before doing so, we should of course apologetically point out that we don't necessarily expect the ideas of the previous section to be at all relevant: both the gluon and graviton are massless, and yet we have assumed thus far that all particle states in our theory of interest are massive [16]. This does not pose a problem for the exercise we are about to carry out, which is simply to quote the known form of amplitudes in the Regge limit, and examine their structure. However, we should keep in mind that when we see results that correlate with the generic Regge behaviour described above (as indeed we will), we should perhaps be surprised!

### 4.1 High energy scattering in QCD

Let us consider $2 \to 2$ scattering of scalar particles that carry a colour charge in a non-abelian gauge theory, where we label momenta according to figure 3(a). If we consider the Regge limit of eq. (61), the momentum transfer between particles 1 and 2 is much smaller than the centre of mass energy. This has a simple physical interpretation, in that it corresponds to highly forward scattering, such that the incoming and outgoing particles suffer a tiny deflection. Each incoming particle does not change species, but instead follows a straight-line trajectory such that the incoming particles (1,2) become collinear with the outgoing particles (3,4) respectively [17]. We show this situation in figure 7, where we note that the two particle trajectories are separated by a transverse distance **b** in general. This is referred to as the *impact parameter*, and would correspond to the distance of closest approach if we were not in the Regge limit. If the (scalar) particles of figure 7 carry colour charge, they can emit gluon radiation, so that we must dress the diagram with all possible virtual gluon emissions, as shown in figure 8. At each order in the strong coupling $g_s$, we can then ask what the dominant terms are in the Regge limit. Remarkably, this can be successfully answered at *all* orders in perturbation theory! Furthermore, the leading terms can be summed up exactly to give a complete function of

---

[16]As noted above, QCD does actually have a mass gap, but this affects larger distances, where the strong interaction will be mediated by mesons rather than the gluon.

[17]This is not the usual *timelike* collinear limit associated with two final state particles becoming mutually collinear. The Regge limit is instead a *spacelike* collinear limit.

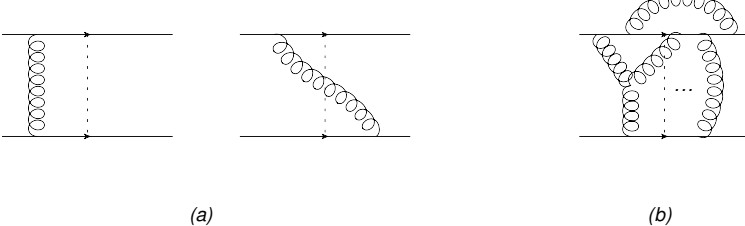

Figure 8: Example virtual gluon radiation at (a) 1-loop order; (b) arbitrary loop order.

the coupling [94, 95] (for related work on this subject, see refs. [96–130]):

$$\mathcal{A}(s,t) \xrightarrow{s \gg -t} \exp\left\{ K \left[ i\pi \mathbf{T}_s^2 + \log\left(\frac{s}{-t}\right) \mathbf{T}_t^2 \right] + \ldots \right\} \mathcal{A}_{\mathrm{LO}}(s,t), \tag{88}$$

where $K$ is a coupling-dependent parameter that we will define shortly, and $\mathcal{A}_{\mathrm{LO}}$ the leading order amplitude for the $2 \to 2$ scattering to take place. The quantities $\mathbf{T}_s^2$ and $\mathbf{T}_t^2$ are operators that act on the colour charge information carried by the LO amplitude. A full definition of them is given in e.g. refs. [131, 132], but we can try to explain the main idea in more qualitative terms here. Imagine that the LO amplitude is dominated by the exchange of a coloured particle (e.g. a gluon) in the $s$-channel. Such amplitudes are eigenstates of the operator $\mathbf{T}_s^2$, and the eigenvalue is just the squared colour charge of the exchanged particle (more formally, the *quadratic Casimir invariant* in the appropriate representation). We can draw this schematically as

$$\mathbf{T}_s^2 \left[ \quad \right] = C_A \qquad , \tag{89}$$

where $C_A$ is indeed the appropriate quadratic Casimir for the gluon. Likewise, eigenstates of the operator $\mathbf{T}_t^2$ are definite exchanges in the $t$-channel, so that we may write

$$\mathbf{T}_t^2 \left[ \quad \right] = C_A \qquad . \tag{90}$$

Imagine that the LO amplitude contains the $t$-channel exchange of a particle $X$, of spin $J$. We already argued in section 3 that this will dominate the physical $s$-channel amplitude in the Regge limit, so that we have

$$\mathcal{A}_{\mathrm{LO}}(s,t) \sim \quad X \quad \sim \left(\frac{s}{-t}\right)^J. \tag{91}$$

Thus, in the Regge limit, the LO amplitude is indeed an eigenstate of the colour operator $\mathbf{T}_t^2$. Furthermore, the exponent in eq. (88) becomes dominated by the second term, which is logarithmically enhanced for $s \gg |t|$. From eqs. (88, 90) we then have

$$\begin{aligned}
\mathcal{A}(s,t) &\to \exp\left\{ K \log\left(\frac{s}{-t}\right) \mathbf{T}_t^2 \right\} \mathcal{A}_{\mathrm{LO}} \\
&= \exp\left\{ K \log\left(\frac{s}{-t}\right) C_X \right\} \mathcal{A}_{\mathrm{LO}} \\
&= \left(\frac{s}{-t}\right)^{K C_X + J}, \tag{92}
\end{aligned}$$

where $C_X$ is the squared colour charge of $X$. The overall effect of all the virtual radiation is as if the propagator for the $X$ particle has been dressed by an overall power of $s/|t|$. For example,

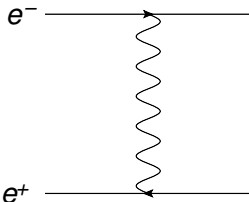

Figure 9: Electron-positron scattering in the high energy limit.

if $X$ is a gluon, we could reproduce the result of eq. (92) by modifying the gluon propagator (e.g. in the Feynman gauge) as follows:

$$-\frac{\eta_{\mu\nu}}{q^2} \to -\frac{\eta_{\mu\nu}}{q^2}\left(\frac{s}{-t}\right)^{KC_A}, \tag{93}$$

where $q$ is the exchanged momentum. This is called *Reggeisation*, and the quantity $KC_A$ is known as the *Regge trajectory* of the gluon. Indeed, we have precisely reproduced the expectations of Regge theory, namely that the physical $s$-channel amplitude in the Regge limit should contain a power-like growth in $s$, where this is associated with particle exchange in the $t$-channel! We could go further than this and show that the Reggeised gluon indeed corresponds to a pole in the complex angular momentum plane, but will not do so.

Note that there are gauge theories, and scattering examples within those theories, such that the squared charge of $t$-channel particles is zero. Consider, for example, the QED scattering process of figure 9, consisting of electron-positron scattering via photon exchange [18]. In this case, the squared charge of the exchanged particle (the photon) is zero, and one can show that the correct abelian form of eq. (88) is

$$\mathcal{A}(s,t) \to e^{i\pi K_{\mathrm{QED}}}\mathcal{A}_{\mathrm{LO}}(s,t), \tag{94}$$

for some parameter $K_{\mathrm{QED}}$ that depends upon the electron charge. Thus, the LO amplitude gets dressed by an overall phase, which is known as the *eikonal phase* in the literature. It is there also in QCD (i.e. as the first term in the exponent of eq. (88)), but in that case is kinematically subleading, in that the Reggeisation term dominates due to the logarithmic enhancement in energy. What is the value of $K$ in eq. (88)? In fact, it turns out to be infinite! To see what has gone wrong, recall that scattering amplitudes contain *infrared (IR) singularities* associated with the emission of virtual radiation which is "soft" (i.e. whose momentum goes to zero). The Regge limit says that the total momentum exchange between the scattering particles must be negligible compared to their energy, which is precisely the statement that the exchanged virtual gauge bosons be soft! We can make $K$ finite by introducing a regulator, and ref. [95] used dimensional regularisation in $d = 4 - 2\epsilon$ dimensions to obtain

$$K = \frac{g_s^2 \Gamma(1-\epsilon)}{4\pi^{2-\epsilon}}\frac{(\mu^2\mathbf{b}^2)^{\epsilon}}{2\epsilon}, \tag{95}$$

where $\mu$ is the dimensional regularisation scale. This depends on the strong coupling $g_s$, as stated above, and also on the impact parameter $\mathbf{b}$, which combines with $\mu$ to make a dimensionless combination. Indeed we see a singularity as $\epsilon \to 0$, corresponding to the IR singularity alluded to above.

---

[18]In figure 9, we can ignore the $s$-channel diagram, which will be kinematically subleading in the Regge limit, at least if we use the Feynman gauge.

Infrared singularities are ultimately a consequence of the fact that interactions involving the exchange of massless particles (e.g. the photon or gluon) are not short-range [19]. Thus, it is not correct to assume that the physical incoming and outgoing states are single particles: they should be dressed with a virtual cloud of gauge bosons. It is possible to set up QFT in this manner in both gauge theories [133,134] and gravity [135], although the procedure is rather cumbersome. Furthermore, in practice one can proceed without this complication. Above, we included only *virtual* radiation. If we included real emissions as well, all IR singularities would cancel [20]. We would then find IR-finite Regge pole effects in the total cross-section. The leading Regge pole with the same quantum numbers as the vacuum is called the *Pomeron*, and has been widely studied (see e.g. refs. [41,136] for a review). We started this section by saying that we should perhaps be surprised if the Regge theory analysis of section 3 (which assumed a mass gap in the theory) turned out to be relevant for theories containing massless particles. We have now seen that it is in fact relevant, provided one can accept the presence of IR singularities. How to interpret the S-matrix in arbitrary massless theories remains a subject of ongoing research.

Interestingly, in the QED case of eq. (94), we can Fourier transform the amplitude from impact parameter space to momentum space exactly, and obtain the result [95]

$$\tilde{A} \sim \left(-\frac{t}{\mu^2}\right)^{-i\alpha} \frac{\Gamma(1+i\alpha)}{\Gamma(1-i\alpha)}, \quad \alpha = \frac{e^2}{4\pi} \frac{s-2m^2}{\sqrt{s(s-4m^2)}}, \tag{96}$$

where $e$ is the electron charge. The gamma function in the numerator has an infinite series of poles, located at

$$s = 2m^2 \left[1 - \left(1 + \frac{e^4}{16\pi^2 N^2}\right)^{-1/2}\right], \quad N \in \mathbb{Z}^+. \tag{97}$$

In section 2, we learnt that poles in the complex $s$ plane should represent bound states, so what are they? The answer is that we have derived (at least in some perturbative approximation) the spectrum of *positronium*, and the above result can be expanded about a non-relativistic limit to obtain known results for the energy levels [137] [21].

## 4.2 High energy scattering in gravity

We can repeat the QCD-like calculation of the previous section in General Relativity (GR), and the answer for the $2 \to 2$ scattering amplitude in the leading Regge limit, dressed with arbitrary amounts of virtual graviton radiation, turns out to be [95]

$$\mathcal{M}(s,\mathbf{b}) \to \exp\left\{K_g\left[i\pi s + t\log\left(\frac{s}{-t}\right)\right]\right\}\mathcal{M}_{\text{LO}}, \tag{98}$$

where $\mathcal{M}_{\text{LO}}$ is the leading-order gravitational amplitude, and

$$K_g = \left(\frac{\kappa}{2}\right)^2 \frac{\Gamma(1-\epsilon)}{4\pi^{2-\epsilon}} \frac{(\mu^2\mathbf{b}^2)^\epsilon}{2\epsilon}. \tag{99}$$

Here $\kappa^2 = 32\pi G_N$, with $G_N$ the Newton constant. Note that eq. (98) is identical in form to the QCD result of eq. (88), and can indeed can be obtained from the latter with the replacements

$$g_s \to \frac{\kappa}{2}, \quad \mathbf{T}_s^2 \to s, \quad \mathbf{T}_t^2 \to t. \tag{100}$$

---

[19]Strong interactions *are* short-range, but this is due to the confinement of the gluon within mesons, which is a non-perturbative effect.

[20]In QCD processes involving incoming quarks / gluons, we would also have to include *parton distribution functions* describing how these are emitted from the incoming hadrons in our experiment. These can then be used to absorb initial state collinear singularities.

[21]Equation (97) has been derived for spinless particles, rather than spin-1/2 electrons and positrons. However, one may show that spin effects are subleading in the Regge limit.

These replacements in fact make perfect sense. The first tells us that the strength of the strong force must be replaced by the appropriate strength of the force in gravity. Furthermore, the operators representing the squared colour charge exchanged in the $s-$ and $t-$channels are replaced with the squared 4-momentum exchanged in these channels. However, in GR the "charge" of a particle (i.e. the thing that makes it couple to the graviton) is its 4-momentum, so that $s$ and $t$ are indeed the squared charges! You may also recognise the replacements of eq. (100) as being consistent with the BCJ double copy of refs. [16–18]. Its appearance here can be understood from the above-mentioned fact that the Regge limit involves the exchange of soft gravitons, and the leading soft structure of QCD and gravity amplitudes is known to double copy [138, 139] (see refs. [140–142] for other studies of the Regge limit in a double copy context).

In eq. (98), we again see the presence of an eikonal phase term [22], and a Reggeisation term, exactly mirroring the structure in the QCD result of eq. (88). However, whereas the Reggeisation term dominated in the QCD case, here the eikonal phase dominates, as it is power-enhanced in $s/|t|$ with respect to the Reggeisation term. Put another way, the Regge trajectory of the graviton is proportional to the squared charge of the graviton, which is $t$ given that it is being exchanged in the $t$-channel. This then means that Reggeisation is a subleading effect compared to the phase, which explains why high energy QCD people talk mainly about Reggeised gluons, whereas high energy gravity people talk mostly about eikonal phases. Both effects are present in both theories, but the physics ends up being very different! There was also some confusion for many years about whether the graviton Reggeises at all (see refs. [143–149] for earlier work), and the above results explain why this confusion persisted for so long: Reggeisation is there, but hidden (by being power-suppressed) in the very limit in which it is expected to occur. As in the QED case, we can Fourier transform to momentum space, and find a spectrum of bound states, as originally discussed in ref. [83]. They are bound states associated with the linearised $(1/r)$ part of the gravitational potential.

Throughout these notes, we have limited ourselves to the case of $2 \to 2$ scattering only. However, Regge theory can also be applied to multiparticle production: see e.g. the classic review of ref. [39]. Indeed, the so-called *multi-Regge limit* has continued to be an important playground for constraining scattering amplitudes in a variety of different field theories [23]. In recent years, a number of studies have examined corrections to the leading Regge limit [85, 86, 150], namely the possibility that the exchanged radiation is no longer strictly soft. People have also applied knowledge from the Regge limit to check and constrain higher-order computations in supergravity [151–155]. There are doubtless many future insights to be gained not only by comparing such calculations with the similar situation in gauge theories (QCD), but also with the vintage physics of the 1960s, which does not assume a particular theory at all! It is hoped that these notes will prove useful in this regard.

# Acknowledgements

I am very grateful to Andreas Brandhuber, Marcel Hughes, Arnau Koemans Collado, Rodolfo Russo and Gabriele Travaglini for useful advice when preparing these lectures. Furthermore, I thank Ben Maybee and Alex Ochirov for encouraging me to make these notes more widely available, and / or for detailed comments on the manuscript.

**Funding information** This work was supported by the UK Science and Technology Facilities Council (STFC) Consolidated Grant ST/P000754/1 "String theory, gauge theory and duality",

---

[22]See refs. [75, 76, 83] for earlier work on the gravitational eikonal phase.

[23]The generalisation of the results of this section to many particle final states was considered in ref. [95].

and by the European Union Horizon 2020 research and innovation programme under the Marie Ckłodowska-Curie grant agreement No. 764850 "SAGEX".

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
