# Peer review of "Aspects of High Energy Scattering"

_SciPost Physics Lecture Notes, doi:SciPost Phys. Lect. Notes 13 (2020)_

## Round 1 · Referee Report · Anonymous (Referee 2) · 2020-1-10

Strengths

This work is pedagogical and well prepared. It is a good submission to the Physics Lecture Notes.

Weaknesses

The work has a fairly narrow scope, and takes a surprisingly olde-fashioned style and tone.

Scientifically, the biggest weakness of this work is common to the field more generally: all of the applications (which are real, and of significant interest) fall explicitly outside the scope of the apparently rigorous introduction---as a massive spectrum of particles is key to the old ideas discussed in section 2. These limitations are discussed in more detail in the report.

Report

In "Aspects of High Energy Scattering", the author gives a clear an pedagogical introduction to Regge theory and `high energy' scattering. There is no doubt that these notes will provide a useful reference to students and researchers (especially less familiar with the intellectual origins of Regge theory).

The work is unquestionably narrow in scope, and unusually olde-fashioned, but neither of these criticisms pose problems for pedagogical works such as this. I would encourage the author do a better job of clarifying the limited scope of this work and its limitations, but I do not view these as critical revisions and leave the decision to the author.

Principal among the limitations that deserve some flagging is the disconnect between the reasonably rigorous treatment of scattering theory where there is an (overly) implicit assumption of massive particles throughout, and the un-discussed leap toward applying these ideas to massless theories (where the arguments in section 2 simply do not apply). To be clear: as this intellectual leap is still a matter of research, it is well beyond the scope of this work to require a bridge from this author. However, it would suit the reader to include a few paragraphs discussing how remarkable (and unjustified) it is that the logic of section 2 may be applied so successfully to the applications of section 4.

Regardless of these limitations, there is no question that this work will be useful to students and researchers alike. I have no doubt that it should be published.

Requested changes

There are no changes that this referee considers mandatory for publication; but there are several areas for improvement that the author may want to consider before publication.

1- I did not notice any reference to Veltman's classic textbook, and the omission seemed odd. The appendix of Diagrammatica at least deserves a mention (if not a thorough comparison to the works cited) in section 2.

2-The author appears to equate "cluster decomposition" with "locality". The author should consider a discussion about how these ideas are related (and inequivalent).

3- In section 2, it would be worth flagging to the reader the role played by m\neq0. Masslessness is more difficult (formally and conceptually), but arguably much more important and interesting.

4-In the discussion at the top of section 2.4: the notation "a(p_1)" is simply confusing and should minimally be clarified. The labels {1,2,3,4} already distinguish the states; the author could write "a(p_a)" etc, or choose some other less redundant and mixed notation; in any case, however, it should be clarified that "a,b,c,d" are being used to signal species.

5-In section 3, the "Regge limit" and "high energy" limit are considered equivalent. The distinction is discussed, but not sufficiently. It is clear that the author wishes to discuss Regge theory---which is simply not the same as "high energy". Indeed,

6-the author may wish to change the title to reflect "Regge Scattering" instead of "High Energy Scattering".

7-In section 3, it is assumed (and insufficiently stated) that all discussion will be about 2->2 scattering. This is indeed interesting, but a wildly strong limitation in scope.

8-The comment about figure 5 not being a Feynman diagram which is in the text, should be included in the caption.

9-Section 4 should start with a discussion and apology that ideas that crucially rely on a mass gap are now going to be applied to a case where the assumptions are not satisfied.

---

## Round 1 · Referee Report · Anonymous (Referee 1) · 2020-1-10

Strengths

The greatest strength of the article is its very clear and focussed approach to discussing the high energy limit and Regge behaviour.

Weaknesses

Considering this as a review article there are no major weaknesses.

Report

This article is a very nicely organised review of well-known arguments about the general structure of scattering amplitudes, based on fundamental principles of quantum mechanics and Lorentz invariance.

Of course further material could be added, but the subject is so large that some cut-off must be applied. For the length of the article as it stands the material covered is impressive.

Requested changes

Here are some very minor comments for improvements:

  1. page 4, 1st paragraph, a typo: "wany" --> "many"

  2. page 4, second paragraph, second sentence: The author refers to "a number" of incoming particles and then assigns them momenta $p_1$ and $p_2$, with ougoing momenta $p_3$ up to $p_n$. This is obviously the case for exactly two incoming particles.

  3. Footnote 2 appears in a strange location, page 4 instead of page 3.

---

## Round 2 · Referee Report · Anonymous · 2020-1-18

Strengths

The revised manuscript shares all of the many strengths of the original submission, and adds many improvements as well.

Report

The work is good, thorough, and extremely pedagogical. It will serve as a useful reference to students and researchers alike. It should be published.

---

## Round 2 · Author Response

Dear colleague,

I am writing to resubmit my lecture notes, following minor revisions at the request of both anonymous referees. My responses to each of them are attached below.

Kind Regards,

Chris White

---

## Round 2 · List of Changes

Response to Report 1:

I thank the referee for their comments, and have made all the requested minor
changes for resubmission.

Response to Report 2:

I thank the referee for their careful reading, and very helpful
suggestions on how to improve the article. A summary of my changes is as follows:

1. I added a reference to Veltman's book in section 2.

2. I have rephrased the discussion of cluster decomposition to remove
this confusion.

3. I have added a discussion at the end of section 2.2 to discuss the
assumption of a mass gap.

4. I have clarified the notation in the caption of figure 3, choosing
to keep letters for particle species, and numbers for 4-momenta. One
reason is that this notation agrees with recent literature on
scattering amplitudes.

5. The referee is strictly speaking correct, although much of the
literature on the Regge limit (including recent studies) uses the terms
"high energy" and "Regge" interchangeably. I have added a clarifying
footnote on p. 19, and hope this does the trick.

6. Please see point 5.

7. The referee is correct about this limitation, which arose due to
the finite length of the lecture notes. I have now added a comment to
address this to the final paragraph of the conclusion, with
references.

8. I have implemented the referee's suggestion.

9. I have added a new introduction to section 4 in line with the
referee's suggestions. Furthermore, I added a clarifying remark at the
end of the first paragraph on p. 30.

You are currently on this page

Resubmission 1909.05177v2 on 17 January 2020

---

## Editorial Decision

published